# Mutagenesis screen uncovers lifespan extension through integrated stress response inhibition without reduced mRNA translation

Maxime J. Derisbourg[1,4], Laura E. Wester[1,4], Ruth Baddi[1] & Martin S. Denzel [1,2,3✉]

Protein homeostasis is modulated by stress response pathways and its deficiency is a hallmark of aging. The integrated stress response (ISR) is a conserved stress-signaling pathway that tunes mRNA translation via phosphorylation of the translation initiation factor eIF2. ISR activation and translation initiation are finely balanced by eIF2 kinases and by the eIF2 guanine nucleotide exchange factor eIF2B. However, the role of the ISR during aging remains poorly understood. Using a genomic mutagenesis screen for longevity in *Caenorhabditis elegans*, we define a role of eIF2 modulation in aging. By inhibiting the ISR, dominant mutations in eIF2B enhance protein homeostasis and increase lifespan. Consistently, full ISR inhibition using phosphorylation-defective eIF2α or pharmacological ISR inhibition prolong lifespan. Lifespan extension through impeding the ISR occurs without a reduction in overall protein synthesis. Instead, we observe changes in the translational efficiency of a subset of mRNAs, of which the putative kinase *kin-35* is required for lifespan extension. Evidently, lifespan is limited by the ISR and its inhibition may provide an intervention in aging.

[1] Max Planck Institute for Biology of Ageing, Cologne, Germany. [2] CECAD - Cluster of Excellence, University of Cologne, Cologne, Germany. [3] Center for Molecular Medicine Cologne (CMMC), University of Cologne, Cologne, Germany. [4]These authors contributed equally: Maxime J. Derisbourg, Laura E. Wester. ✉email: martin.denzel@age.mpg.de

A ging is defined as the progressive loss of physiological integrity accompanied by reduced cellular, organ, and systemic performance. It is characterized by cellular hallmarks, such as stem cell exhaustion, genomic instability, deregulated nutrient sensing, and loss of protein homeostasis[1]. Thus, aging is the main risk factor for neurodegenerative disorders, cancer, and metabolic syndrome. The aging process can be modulated by environmental and genetic factors, and several evolutionarily conserved biological processes have been implicated in lifespan regulation[2]. Numerous longevity pathways have been defined using forward genetic screens in the nematode *Caenorhabditis elegans* and *age-1*/PI3-kinase was first identified as a longevity gene in a chemical mutagenesis screen[3,4]. Many longevity genes in conserved pathways were then identified and further illuminated by genomic RNA interference (RNAi) screens[5–7]. RNAi, however, remains limited as it leads to varying degrees of mRNA knockdown, and it does not have the resolution to investigate consequences of other genetic alterations, including gain-of-function mutations. Importantly, functions of essential genes cannot be investigated by RNAi. Point mutagenesis, in contrast, can give valuable insight into functions of essential genes, for example through separation-of-function mutations[8,9]. The chemically induced nucleotide changes in random mutagenesis can lead to multiple types of mutations, including single amino acid substitutions and gain-of-function mutations[10], thus reaching unparalleled resolution. Despite the high resolution of random chemical mutagenesis in defining relevant phenotypes and the analytical power of genome sequencing, an unbiased forward longevity screen using chemical mutagenesis coupled with whole-genome sequencing has not been done to date.

Failure of protein homeostasis occurs early during aging and various interventions that promote or maintain protein homeostasis beneficially affect lifespan in model organisms[10–12]. Maintenance of protein homeostasis by cellular stress response pathways is an essential feature of cellular integrity and organismal resilience. Internal and external stimuli trigger evolutionarily conserved cellular stress pathways such as the heat shock response, organelle-specific stress response pathways including the endoplasmic reticulum or mitochondrial unfolded protein responses (ER-UPR/mito-UPR), and the integrated stress response (ISR). Multiple lines of evidence show that longevity ultimately relies on the fidelity of cellular stress response mechanisms[13].

The biological function of the ISR is to restore cellular homeostasis upon stress. In mammals, the activation of the ISR relies on the eukaryotic initiation factor 2 (eIF2) kinases: heme-regulated inhibitor (HRI), protein kinase R (PKR), general control nonderepressible 2 (GCN2), and PKR-like endoplasmic reticulum kinase (PERK). They are activated, respectively, by iron deficiency, viral infection, amino acid deprivation, and accumulation of misfolded proteins in the ER[14]. In *C. elegans*, GCN-2 and PERK/PEK-1 are the eIF2 kinases. The kinases converge on the phosphorylation of the α subunit of eIF2, which is a key regulator of translation initiation, the limiting step of protein synthesis[15]. For translation initiation to occur, the eIF2.GTP. tRNA$^{met}$ ternary complex together with other initiation factors and the 40S ribosomal subunit form the 43S pre-initiation complex. The 43S complex binds to the 5′-cap structure and scans along the mRNA until it recognizes the AUG start codon. Then, GTP hydrolysis releases eIF2 and other initiation factors from the mRNA–40S-complex, allowing the 60S ribosomal subunit to bind and proceed to elongation[16]. The exchange of GDP to GTP is necessary for recycling eIF2 back to its active form and for further rounds of translation initiation. This exchange is catalyzed by the heterodecameric guanine nucleotide exchange factor eIF2B[17]. The phosphorylation of eIF2α at serine 51 by the stress-sensitive

kinases represents the core event of the ISR. Phospho-eIF2 is a strong inhibitor of eIF2B leading to attenuated ternary complex formation and therefore to a reduction of 5′-cap-dependent protein synthesis[18,19]. Decreasing ternary complex abundance paradoxically derepresses translation of specific mRNAs that are regulated by upstream open reading frames (uORFs), for example, ATF-4[20,21]. While the ISR and mRNA translation initiation are finely balanced to provide robustness during acute challenges to protein homeostasis, the role of this pathway during aging and in longevity remains largely unexplored.

Here, we set out to perform a large-scale mutagenesis screen for increased survival in *C. elegans*. We sequenced the genomes of over 100 long-lived mutant strains and identified a convergence of multiple independent causal longevity loci in regulators of eIF2. Importantly, lifespan extension and resistance to proteotoxicity occurred without repressing overall protein biosynthesis. Instead, we observed the selective regulation in the translation of a subset of mRNAs. Among these, the putative kinase *kin-35* was required for lifespan extension. Further analysis revealed that the lifespan-extending mutations of eIF2 regulators in fact were ISR inhibitors. We found that pharmacological ISR inhibition, as well as genetic ISR ablation by mutating serine 51 of eIF2α likewise results in lifespan extension. Together, we demonstrate that ISR inhibition enhances protein homeostasis and extends survival without inhibiting protein synthesis.

## Results

**Multiple genes controlling mRNA translation initiation affect longevity and protein homeostasis in *C. elegans*.** To identify modulators of the aging process, we performed an unbiased forward longevity genetic screen[3,4] that combines chemical mutagenesis with deep sequencing (Fig. 1a). The conditionally sterile CF512 strain [*fer-15(b26)*; *fem-1(hc17)*] was mutagenized with 0.3% ethyl methanesulfonate (EMS). 28,000 tested genomes were screened for maximum lifespan extension after growth at nonpermissive temperature until the L4 larval stage to induce sterility. 283 mutant strains showed increased maximum lifespan and after full demographic analysis we sequenced 101 genomes of mutants with a mean lifespan extension of at least 18% (Fig. 1b). Validating the approach, we identified mutations in the insulin signaling pathway whose disruption is known to increase lifespan[22]. We found six new alleles of the *daf-2* insulin/insulin-like growth factor 1 (IGF-1) receptor gene (Supplementary Fig. 1a, b). A specific phenotype of *daf-2* mutant larvae is a reduced threshold for entering the developmental dauer state[23]. To test if the *daf-2* alleles might indeed be linked to altered insulin signaling, we quantified heat-induced dauer formation. We observed enhanced dauer formation in three of the six *daf-2* mutant strains (Supplementary Fig. 1c). We further analyzed dauer alae, a dauer-specific morphologic trait of the worm cuticle. While CF512 control animals developed regular cuticle alae when grown at 27 °C, the three dauer-constitutive mutants also developed dauer alae (Supplementary Fig. 1d). These results suggest that reduced insulin signaling pathway activity might extend lifespan in the *daf-2* mutants found in the screen. In addition to point mutations in *daf-2*, we found four uncharacterized alleles of *che-3* and one of *osm-3* whose mutations disrupt chemosensation, extending lifespan[24]. In addition, we identified two mutations in *ifg-1* and two mutations in *ife-2*, genes linked to lifespan extension through reduced mRNA translation[25,26] (Supplementary Fig. 1e). Taken together, these observations validate the screening approach that was expected to reconfirm known longevity pathways.

Besides known longevity genes, our genomic analysis revealed a cluster of alleles in genes that control the initiation step of mRNA translation. We found two independent alleles in *ppp-1*/eIF2Bγ, one

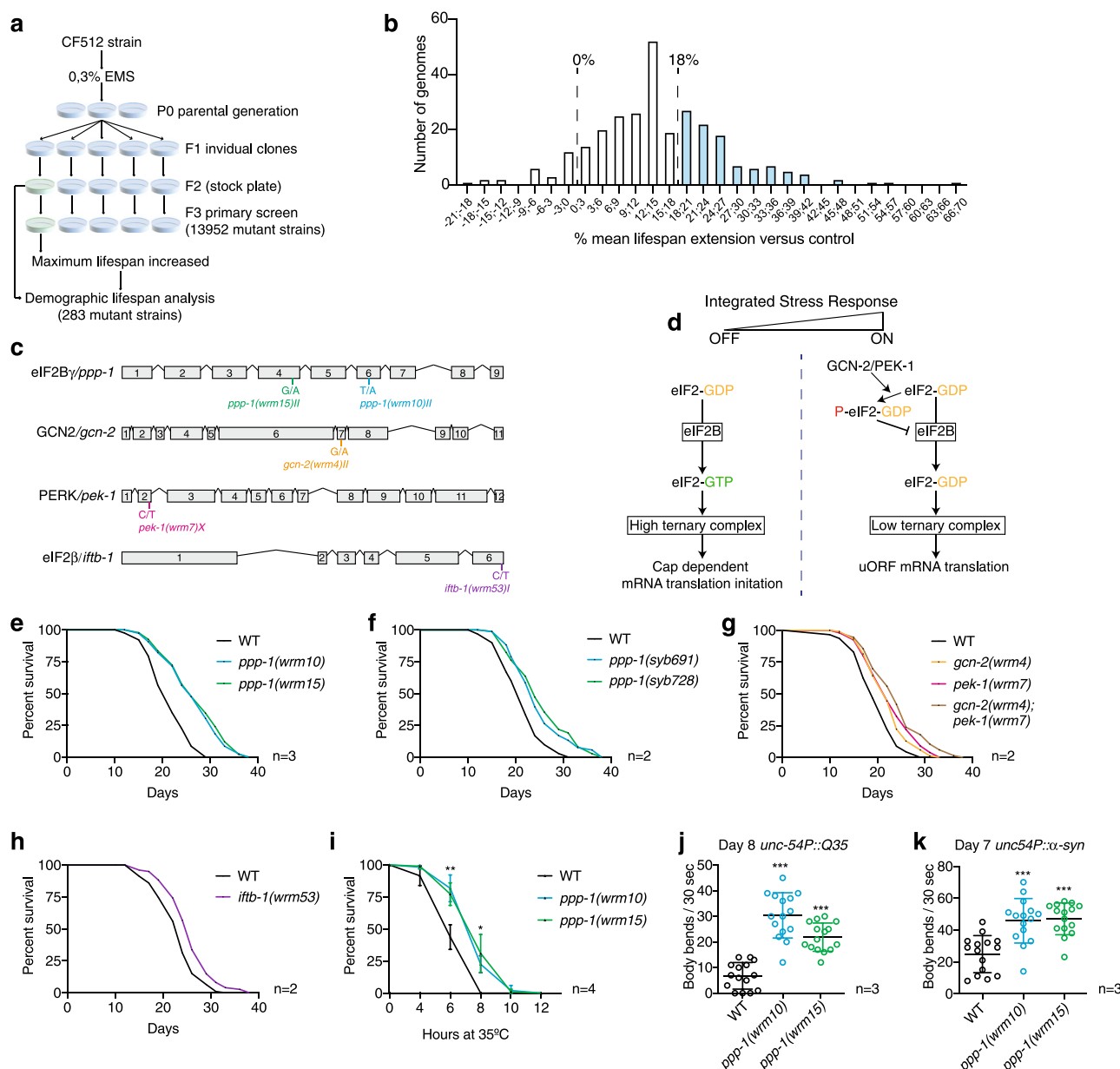

**Fig. 1 Unbiased forward longevity screen in *C. elegans* identifies mutations in ISR components. a** Mutagenesis screening strategy. **b** Mean lifespan extension (normalized to temperature-sensitive sterile CF512 control) visualized as the number of tested genomes in 3% bins. **c** Schematic representation of identified ISR genes and corresponding longevity alleles. **d** Cartoon depiction of mRNA translation initiation and the ISR. **e** Survival of outcrossed *ppp-1* (*wrm10*) and *ppp-1*(*wrm15*) mutants compared to WT controls (representative data from $n = 3$ independent experiments). **f** Survival of CRISPR/Cas9-generated *ppp-1* alleles *syb691* and *syb728* compared to WT controls (representative data from $n = 2$ independent experiments). *syb691* corresponds to *wrm10* and *syb728* to *wrm15*. **g** Survival of outcrossed *gcn-2*(*wrm4*), *pek-1*(*wrm7*), and *gcn-2*(*wrm4*); *pek-1*(*wrm7*) double mutants compared to WT controls (representative data from $n = 2$ independent experiments). **h** Survival of outcrossed *iftb-1*(*wrm53*) mutants compared to WT controls (representative data from $n = 2$ independent experiments). **i** Thermotolerance assays of day 1 *ppp-1* mutant worms compared to WT controls (error bars represent means ± SD, two-way ANOVA Dunnett's post hoc test with **$p = 0.0032$ *ppp-1*(*wrm10*) vs. WT at the 6 h time point; **$p = 0.0039$ *ppp-1*(*wrm15*) vs WT at the 6 h time point; *$p = 0.0107$ *ppp-1*(*wrm10*) vs. WT at the 8 h time point; *$p = 0.0404$ *ppp-1*(*wrm15*) vs. WT at the 8 h time point; $n = 4$ independent experiments with 50 animals each). **j** Motility assay using day 8 WT animals and *ppp-1* mutants with an *unc-54P*-driven muscle-specific expression of polyQ35–YFP fusion protein (error bars represent means ± SD, one-way ANOVA Dunnett's post hoc test with ***$p < 0.001$ vs. WT controls; $n = 3$ independent experiments with ≥12 animals each). **k** Motility assay using day 7 WT animals and *ppp-1* mutants with an *unc-54P*-driven muscle-specific expression of α-synuclein (error bars represent means ± SD, one-way ANOVA Dunnett's post hoc test with ***$p < 0.001$ vs. WT controls; $n = 3$ independent experiments with 15 worms each). See Supplementary Dataset 1 for survival statistics. See Supplementary Table 1 for statistics on thermotolerance assays. Source data are provided as a Source Data file.

mutation in *gcn-2*/GCN-2, one mutation in *pek-1*/PERK and one mutation in *iftb-1*/eIF2β (Fig. 1c; Supplementary Fig. 1f). These results suggest a link between ISR regulation (Fig. 1d) and *C. elegans* longevity. To reduce the background mutational load, we outcrossed

the *ppp-1*(*wrm10*) and *ppp-1*(*wrm15*) alleles and found that they extend *C. elegans* lifespan by 20% (Fig. 1e and Supplementary Dataset 1). Furthermore, CRISPR/Cas9 generated mutants with identical substitutions confirmed the longevity (Fig. 1f). The

outcrossed *gcn-2(wrm4)* and *pek-1(wrm7)* mutants as well as the *gcn-2(wrm4); pek-1(wrm7)* double mutant were long-lived (Fig. 1g). Finally, the outcrossed *iftb-1(wrm53)* mutant strain also showed a mild lifespan extension (Fig. 1h). Taken together, we identified causal mutations for longevity from the chemical screen and found a link between the regulation of the ISR and longevity.

Given their robust lifespan extension and the role of protein homeostasis in other longevity pathways, we further characterized *ppp-1* mutants using proteotoxic challenges. Upon heat shock, *ppp-1* mutants showed enhanced survival compared to wild type (WT) animals (Fig. 1i; Supplementary Table 1). Expression of fluorescently tagged polyglutamine (polyQ35) in the muscle[27] results in a drastic decrease of motility (Fig. 1j). Strikingly, *ppp-1* mutants were protected from polyQ35 toxicity (Fig. 1j). Similarly, *ppp-1* mutations reduced the paralysis in a transgenic strain expressing fluorescently tagged α-synuclein[28] (Fig. 1k). In sum, the chemical mutagenesis screen, a fully unbiased genome-wide approach to find longevity loci in *C. elegans*, identified *ppp-1* point mutations that lead to proteotoxic stress resistance and extended survival.

**ppp-1 longevity is independent of attenuated translation.** Reduction of protein synthesis by partially or fully abolishing the activity of translation initiation or elongation factors can result in lifespan extension[25,29–31]. As eIF2B and eIF2 are direct key regulators of mRNA translation initiation, we monitored protein synthesis in *ppp-1* and *iftb-1* mutants. We used surface sensing of translation (SUnSET) to measure protein synthesis rates. This technique is based on the incorporation of puromycin into newly synthesized proteins, followed by their detection with a monoclonal antibody[32]. We found reduced puromycin incorporation in *iftb-1* mutants, suggesting that they have reduced protein synthesis (Fig. 2a). This is consistent with published data that link attenuated translation after *iftb-1* RNAi treatment to longevity[25]. Next, we monitored protein synthesis in the *ppp-1* mutants using SUnSET. Surprisingly, no changes in protein synthesis were observed between *ppp-1* mutants and WT animals whereas control *rsks-1*/S6K mutants showed a drastic reduction of puromycin-labeled peptides (Fig. 2b). Verifying these results, we used radioactive methionine incorporation to measure mRNA translation and did not observe any differences between WT animals and *ppp-1* mutants, while *rsks-1*/S6K mutants showed a drastic reduction (Fig. 2c). To gain a deeper understanding of potential changes of mRNA translation and to evaluate ribosomal activity in *ppp-1* mutants, we performed polysome profiling. We found no differences in the overall ribosome distribution in *ppp-1* mutants compared to WT animals (Fig. 2d, e). To test if the *gcn-2 (wrm4); pek-1(wrm7)* double mutation might affect survival through reducing protein synthesis, we likewise analyzed ribosomal activity. Polysome profiles demonstrated that the *gcn-2 (wrm4); pek-1(wrm7)* mutants do not have altered overall translation (Supplementary Fig. 2a, b). Thus, the longevity screen unraveled two distinct classes of mutants among the eIF2 regulators: first, a mutation in *iftb-1* reducing protein synthesis and, second, longevity-associated mutations that do not affect bulk protein synthesis such as in *ppp-1*.

Since eIF2 activity is regulated by phosphorylation, we also evaluated phospho-eIF2α on day 1 and day 6 of adulthood. We found that the phosphorylation of eIF2α was increased in aged WT animals (Supplementary Fig. 2c). However, we did not observe any differences between *ppp-1* mutants and the WT control at day 1 or day 6. Together, our results support the idea that improved protein homeostasis and longevity of *ppp-1* mutants are uncoupled from reduced protein synthesis.

**kin-35 translation is required for ppp-1 longevity.** As we did not observe any changes in global protein synthesis, we asked whether the translational efficiency of specific mRNAs might extend the lifespan of the *ppp-1* mutant animals. Since polysome association is indicative of higher translation of mRNAs, we compared the ratio of polysome-associated mRNAs (>3 ribosomes/mRNA) normalized to total mRNA levels between WT and *ppp-1* animals (Fig. 3a). We found a significant de-enrichment of 336 mRNAs and an enrichment of 72 mRNAs in the polysome fractions of *ppp-1* mutants (Fig. 3b and Supplementary Dataset 2). GO term analysis of all significantly changed polysome-associated mRNAs revealed enrichment for genes involved in phosphorylation (Fig. 3c).

We hypothesized that some of the enriched mRNAs might contribute to longevity and define *ppp-1* phenotypes. We used resistance to polyQ35 proteotoxicity of *ppp-1* animals as a proxy for longevity and individually knocked down the candidate mRNAs in *ppp-1(wrm10)* mutants using RNAi. At day 8 of adulthood, all polyQ35 transgenic animals were paralyzed while polyQ35; *ppp-1(wrm10)* animals remained motile. Thus, we screened for suppressors of the paralysis phenotype in *ppp-1* mutants (Fig. 3d; Supplementary Fig. 3a). Knockdown of 7 mRNAs suppressed *ppp-1* motility by at least 50% (Fig. 3d, in yellow). Further validation of these 7 RNAi clones was performed by quantifying motility in liquid in both *ppp-1* mutants. Knockdown of candidate genes C01A2.5 and M04F3.3 showed significant motility reduction in both *ppp-1* mutants (Fig. 3e) without affecting motility in WT animals (Supplementary Fig. 3b). These two clones were selected for lifespan analysis and knockdown of M04F3.3 showed full suppression of *ppp-1* longevity without limiting WT lifespan (Fig. 3f), and knockdown of M04F3.3 mRNA was shown to be effective (Supplementary Fig. 3c). M04F3.3 encodes a predicted kinase with yet unknown functions in the worm that we termed *kin-35*. qPCR analysis confirmed that *kin-35* mRNA association with polysomes was enhanced in *ppp-1* mutants without increased allover abundance (Supplementary Fig. 3d). Together, these data suggest that increased translation of *kin-35* mRNA is required for *ppp-1* longevity. C01A2.5 knockdown also significantly reduced *ppp-1* longevity but shortened WT lifespan suggesting general toxicity (Supplementary Fig. 3e).

We next asked if *kin-35* over-expression might be sufficient to extend lifespan in WT animals. We generated three independent *kin-35* over-expressing lines (Supplementary Fig. 3f). We did not observe any lifespan extension (Fig. 3g; Supplementary Fig. 3g) or increased thermotolerance (Fig. 3h; Supplementary Fig. 3h) in the *kin-35* transgenic lines. Overall, our results demonstrate that selective translation of *kin-35* is required for lifespan extension and increased protein homeostasis in *ppp-1* animals. However, *kin-35* over-expression was not sufficient to extend lifespan and bolster protein homeostasis in WT nematodes.

**ppp-1 mutations inhibit the ISR.** To further characterize the functional relevance of the *ppp-1* mutations, we used RNAi-mediated *ppp-1* silencing. Knockdown of *ppp-1* did not affect the survival of WT animals (Fig. 4a). Instead, *ppp-1* RNAi abolished longevity and heat resistance of both *ppp-1* mutants (Fig. 4a; Supplementary Fig. 4a) and heterozygous *ppp-1* mutants were long-lived (Fig. 4b). These observations exclude the possibility of causal loss-of-function *ppp-1* mutations as eIF2B activity was required for the observed longevity phenotype. The data, therefore, suggest that *ppp-1* mutations are genetically dominant. Activation of *ppp-1*, hence of the eIF2B complex, would counter the effect of eIF2α phosphorylation and blunt the ISR upon stress. To test this hypothesis, we monitored the uORF-regulated

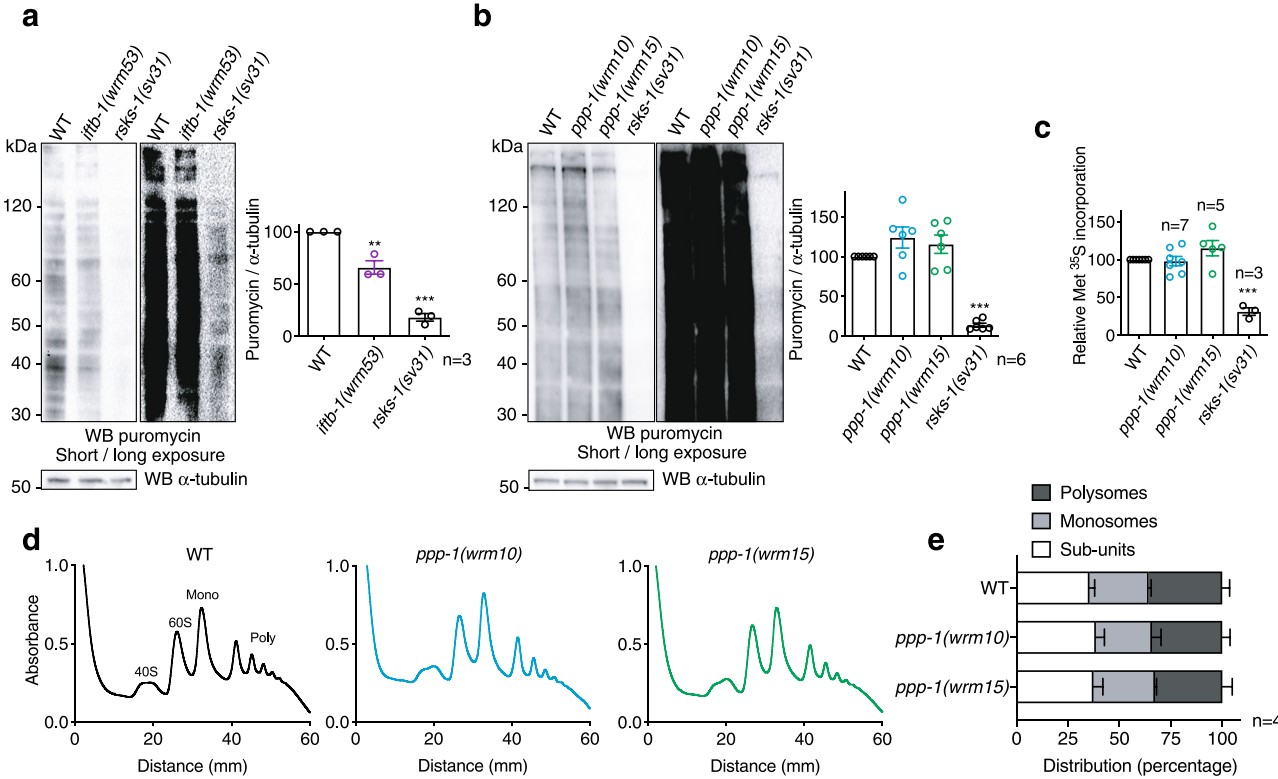

**Fig. 2 Mutations in *ppp-1* do not attenuate mRNA translation. a** Puromycin incorporation followed by Western blot analysis using antibodies detecting puromycin and α-tubulin in day 1 WT animals, *iftb-1(wrm53)* mutants, and control *rsks-1(sv31)* mutants. Images of the same membrane are shown with short and long exposures (error bars represent means + SEM, one-way ANOVA Dunnett's post hoc test with **$p = 0.0023$ *iftb-1(wrm53)* vs. WT and ***$p < 0.001$ *rsks-1(sv31)* vs. WT; $n = 3$ independent experiments). **b** Puromycin incorporation assay in day 1 WT animals, *ppp-1* mutants, and *rsks-1(sv31)* controls as described in (**a**). Images of the same membrane are shown with short and long exposures (error bars represent means + SEM, one-way ANOVA Dunnett's post hoc test with ***$p < 0.001$ vs. WT; $n = 6$ independent experiments). **c** Quantification of methionine $^{35}$S labeling of day 1 WT worms, *ppp-1* mutants, and control *rsks-1(sv31)* mutants (error bars represent means + SEM, one-way ANOVA Dunnett's post hoc test with ***$p < 0.001$ vs. WT; the number of independent experiments ($n$) indicated in the figure). **d**, **e** Polysome profiling, and quantification of day 1 WT and *ppp-1* animals. Quantification represents the relative abundance of ribosomal subunits (40S, 60S), monosomes (mono), and polysomes (poly; error bars represent means + SD, two-way ANOVA Dunnett's post hoc test; $n = 4$ independent experiments). Source data are provided as a Source Data file.

translational activation of the worm homolog of GCN4/ATF4, *atf-4* in the translational *atf-4::GFP* reporter strain. *C. elegans atf-4* was previously named *atf-5*[33]. We used tunicamycin, a specific ER stress inducer that perturbs N-glycosylation, at varying doses to induce reporter expression. Interestingly, we observed that at high tunicamycin concentrations, WT animals reached a plateau of GFP intensity, suggesting that the ISR reached its maximum (Fig. 4c, d). Upon tunicamycin administration, both *ppp-1* mutants showed a significant reduction of the GFP signal compared to WT controls, demonstrating a blunted ISR even at higher tunicamycin concentrations (Fig. 4c, d). We further validated these results with DTT that induces ER stress by interfering with disulfide bond formation and thus protein maturation. While DTT treatment significantly increased reporter expression in WT animals, both *ppp-1* alleles blunted the *atf-4::GFP* response (Supplementary Fig. 4b, c). As *ppp-1* mutations inhibit the ISR during stress, we wondered whether other stress pathways were regulated and therefore investigated the state of the ER-UPR in *ppp-1* mutants. We monitored both constitutive (Supplementary Fig. 4d) and inducible UPR (Supplementary Fig. 4e) target genes. We did not observe any increase of UPR activity in *ppp-1* mutants. Further, we asked if the stress resistance of *ppp-1* mutants might be linked to changes in insulin signaling. We did not observe elevated heat-induced dauer formation and did not detect any changes in the expression of *daf-16*/FOXO target genes in *ppp-1* mutants (Supplementary Fig. 4f, g). Together, these data

support a specific ISR effect of the *ppp-1* mutations without compensatory regulation of the UPR or adaptive changes in the insulin signaling pathway.

As we observed a specific effect of the *ppp-1* mutants in the ISR, we next tested whether the *gcn-2(wrm4)* and *pek-1(wrm7)* mutations from the longevity screen might also prevent ISR activation. Since *C. elegans* has only two eIF2 kinases, GCN-2, and PEK-1, we were able to separate the *wrm4* and *wrm7* alleles from the respective other kinase. We generated double mutants using the *pek-1(ok275)* and *gcn-2(ok871)* full knockout mutants. The *gcn-2(wrm4); pek-1(ok275)* double mutant only has the remaining GCN-2 activity and displayed an 80% reduction of baseline eIF2α phosphorylation (Supplementary Fig. 4h). Likewise, the remaining PEK-1 activity in the *pek-1(wrm7); gcn-2 (ok871)* double mutant was significantly reduced (Supplementary Fig. 4h). We conclude that the single amino acid substitutions in GCN-2 and PEK-1 reduce their baseline activity in the ISR.

**Pharmacological ISR inhibition promotes longevity.** To further validate our findings using genetic ISR modulators, we next asked if pharmacological ISR inhibition might affect survival in *C. elegans*. For this, we used a set of compounds that were previously described as UPR modulators in worms[34]. We demonstrated that estradiol valerate is an ISR inhibitor, as it reduced GFP induction of the *atf-4::GFP* reporter during tunicamycin treatment (Fig. 5a). Consistent with ISR inhibition, estradiol valerate did not suppress

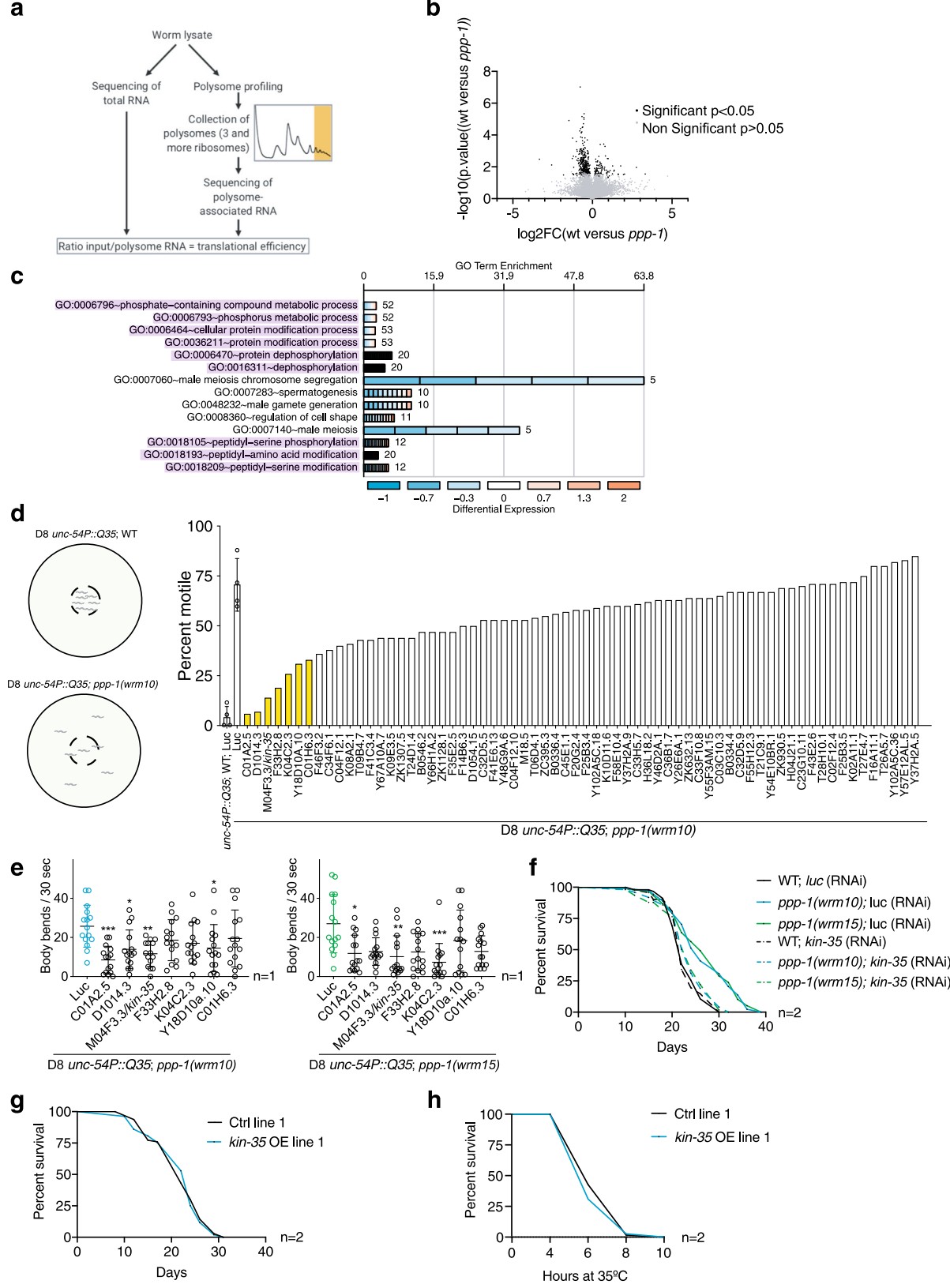

overall protein biosynthesis (Supplementary Fig. 5a). ISRIB, a well-understood inhibitor of the mammalian ISR[35,36], did not affect *atf-4::GFP* levels in *C. elegans* and did not affect survival (Fig. 5a and Supplementary Fig. 5b). Propafenone hydrochloride further elevated GFP expression, showing that it is an ISR activator (Fig. 5a). Estradiol valerate significantly extended *C. elegans* lifespan (Fig. 5b)

and suppressed eIF2α phosphorylation upon DTT treatment (Fig. 5c). Surprisingly, estradiol valerate treatment initiated at day 5 or day 10 of adulthood equally increased survival (Fig. 5b) suggesting that late ISR inhibition might be sufficient to promote lifespan extension. Other estradiol derivatives do not interfere with the ISR[34], suggesting specificity. In agreement with the genetic

**Fig. 3 Translational efficiency is altered in *ppp-1* mutants. a** Polysome sequencing strategy. **b** Volcano plot of polysome-associated mRNAs normalized to total mRNA levels between WT animals and *ppp-1* mutants. All displayed mRNAs were found in both *ppp-1* mutants. Mean p-values and mean log-2-fold changes of both *ppp-1* mutants were used (Student's two-sided *t* test, significance is reached for $p < 0.05$). FC fold change. The full dataset is in Supplementary Dataset 2. **c** DAVID gene ontology (GO) analysis of significantly changed mRNAs shown in (**b**). Processes involved in phosphorylation are highlighted in purple. **d** RNAi screen for suppressors of polyQ35; *ppp-1(wrm10)* motility. For reliability, assays of polyQ35 WT animals and polyQ35; *ppp-1(wrm10)* mutants on *luciferase* RNAi were performed four times (error bars represent means + SD). Bars highlighted in yellow indicate RNAi treatments with a reduction of motility of at least 50% compared to the *luciferase* control treatment of polyQ35; *ppp-1(wrm10)* worms. **e** Motility assays of day 8 polyQ35 transgenic animals, polyQ35; *ppp-1(wrm10)* and polyQ35; *ppp-1(wrm15)* mutants after indicated RNAi treatments (error bars represent means ± SD, one-way ANOVA Kruskal Wallis test with ***$p < 0.001$ C01A2.5 vs. Luc, *$p = 0.0282$ D1014.3 vs. Luc, **$p = 0.0074$ M04F3.3/*kin-35* vs. Luc, and *$p = 0.0361$ Y18D10a.10 vs. Luc in polyQ35; *ppp-1(wrm10)* animals (left panel) and *$p = 0.0397$ C01A2.5 vs. Luc, **$p = 0.0036$ M04F3.3/*kin-35* vs. Luc, and ***$p < 0.001$ K04C2.3 vs. Luc in polyQ35; *ppp-1(wrm15)* animals (right panel); $n = 1$ with ≥14 animals per treatment). **f** Survival of WT animals and *ppp-1* mutants upon M04F3.3/*kin-35* or control *luciferase* RNAi knockdown (representative data from $n = 2$ independent experiments). **g** Survival of *kin-35* over-expressing worms (line 1) compared to respective control animals without the extrachromosomal array (representative data from $n = 2$ independent experiments). **h** Thermotolerance assay of day 1 *kin-35* over-expressing mutants (line 1) compared to respective controls as described in (**g**) (representative data from $n = 2$ independent experiments with ≥30 worms each). OE overexpressor. See Supplementary Dataset 1 for survival statistics. See Supplementary Table 1 for statistics on thermotolerance assays. Source data are provided as a Source Data file.

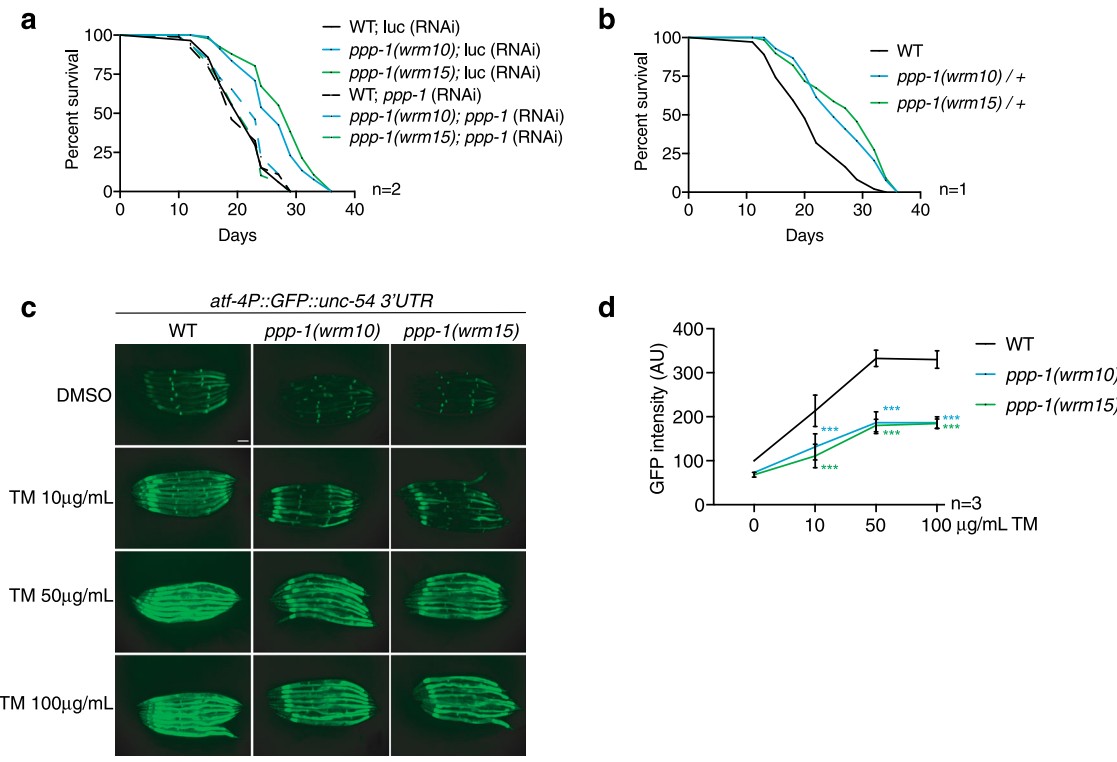

**Fig. 4 Mutations in *ppp-1* reduce the ISR upon stress. a** Survival of WT animals and *ppp-1* mutants upon RNAi treatment targeting *ppp-1* or control *luciferase* (representative data from $n = 2$ independent experiments). **b** Survival of heterozygous *ppp-1* mutants compared to WT control animals. **c** Fluorescence images of day 1 WT animals and *ppp-1* mutants in the *atf-4P*::GFP::*unc-54* 3'UTR reporter background. Worms were treated with DMSO only (control) or with the indicated tunicamycin (TM) concentrations for 6 h each (scale bar 75 μm; representative data from $n = 3$ independent experiments with ≥7 animals each). **d** Quantification of GFP intensity in (**c**). Error bars represent means ± SD, two-way ANOVA Dunnett's post hoc test with ***$p < 0.001$ vs. WT ($n = 3$ independent experiments with ≥7 animals each). See Supplementary Dataset 1 for survival statistics. Source data are provided as a Source Data file.

interaction of *ppp-1* and *kin-35*, lifespan extension mediated by estradiol valerate was *kin-35* dependent as *kin-35* silencing abolished the longevity specifically in treated animals while not affecting untreated controls (Fig. 5d). Furthermore, lifespan extension by estradiol valerate treatment or *ppp-1* mutation were not additive, supporting the role of estradiol valerate in ISR modulation (Fig. 5e). Finally, ISR induction with propafenone hydrochloride shortened lifespan (Supplementary Fig. 5c). Taken together, our results demonstrate that pharmacological ISR activation was detrimental while ISR inhibition, even late in life, extended lifespan.

**Disabling eIF2 phosphorylation extends lifespan in *C. elegans*.** To mechanistically address whether ISR inhibition leads to longevity, we engineered a phospho-defective *eIF2αS51A* mutant [Y37E3.10a(*syb1385*)], abolishing the key molecular event in the ISR (Fig. 6a). Homozygous *eIF2αS51A* mutants were viable and displayed regular pharyngeal pumping rates, generation time, and brood size (Supplementary Fig. 6a–c). Importantly, during development *eIF2αS51A* mutants were hypersensitive to ER stress induced by tunicamycin, likely because phosphorylation of eIF2α by the *pek-1*/PERK kinase is required to promote the ER stress response and survival (Fig. 6b). Notably, *eIF2αS51A* mutants

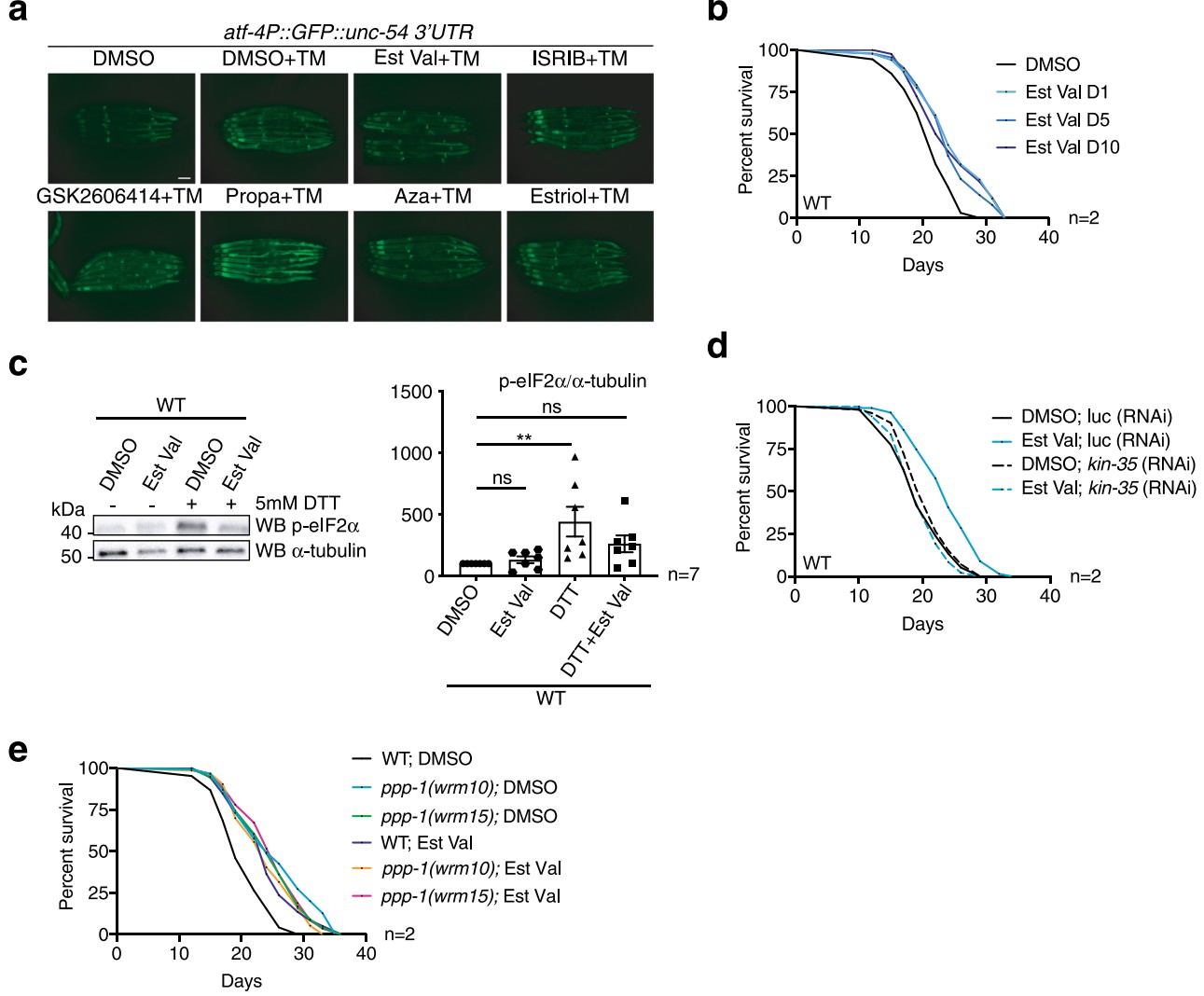

**Fig. 5 Estradiol valerate inhibits the ISR and extends lifespan. a** Fluorescence images of *atf-4P*::GFP::*unc-54* 3'UTR reporter animals grown on NGM plates supplemented with 10 μg/mL tunicamycin (TM) and with the indicated compounds (20 μM) or 1% DMSO vehicle control (Est Val = estradiol valerate, Propa = propafenone hydrochloride, Aza = azadirachtin). Scale bar is 75 μm, $n = 1$. **b** Survival of WT worms treated with 1% DMSO (control) or 20 μM estradiol valerate from day 1 (D1), day 5 (D5), or day 10 (D10) (representative data from $n = 2$ independent experiments). **c** Representative Western blot of day 1 worms treated with 1% DMSO (control) or 20 μM estradiol valerate. Worms were incubated without (−) or with 5 mM DTT (+) for 2 h. Levels of phospho-eIF2α (Ser51) were normalized to α-tubulin (error bars represent means +SEM, one-way ANOVA Dunnett's post hoc test, **$p = 0.0044$ DTT vs. DMSO; ns = not significant vs. DMSO; $n = 7$ independent experiments). **d** Survival of WT worms treated with 1% DMSO (control) or 20 μM estradiol valerate from day 1 of adulthood upon RNAi treatment targeting *kin-35* or control *luciferase* (representative data from $n = 2$ independent experiments). **e** Survival of *ppp-1* mutants and WT controls treated with 1% DMSO (control) or 20 μM estradiol valerate from day 1 of adulthood (representative data from $n = 2$ independent experiments). See Supplementary Dataset 1 for survival statistics. Source data are provided as a Source Data file.

showed a robust increase in survival compared to WT animals, demonstrating that the genetic inhibition of the ISR extends lifespan (Fig. 6c). We next asked if, in a trade-off with the lifespan extension, genetic inhibition of the ISR might render worms hypersensitive to chronic stress and performed survival assays on tunicamycin. As expected, WT lifespan was reduced upon chronic ER stress and eIF2αS51A mutants had the same lifespan as WT animals in the presence of tunicamycin (Supplementary Fig. 6d). We conclude that genetic ablation of the ISR does not render worms hypersensitive to tunicamycin in adulthood. Consistent with the *ppp-1* thermotolerance phenotype (Fig. 1i), eIF2αS51A mutants were heat resistant (Fig. 6d). Furthermore, polysome profiling and puromycin incorporation revealed that the lifespan extension of eIF2αS51A mutants occurred without a

reduction in protein synthesis (Fig. 6e, f, Supplementary Fig. 6e). In fact, the longevity of the eIF2αS51A mutant was dependent on *kin-35* as *kin-35* RNAi treatment abolished the longevity of the ISR defective mutant (Fig. 6g). This suggests that the lifespan extension mediated by ISR inhibition is dependent on selective translation, similar to the *ppp-1* mutants. Finally, we investigated the genetic interaction between *ppp-1* and eIF2α and asked if eIF2B was required for eIF2αS51A longevity. Strikingly, *ppp-1* silencing completely suppressed the longevity of eIF2αS51A mutants, suggesting that the eIF2B complex mediates longevity in the ISR ablated mutant (Fig. 6h).

We next tested if, in turn, genetically activated ISR signaling would display detrimental phenotypes. To this end, we generated a phosphomimic eIF2αS51D mutant [Y37E3.10a*(syb1567)*] using

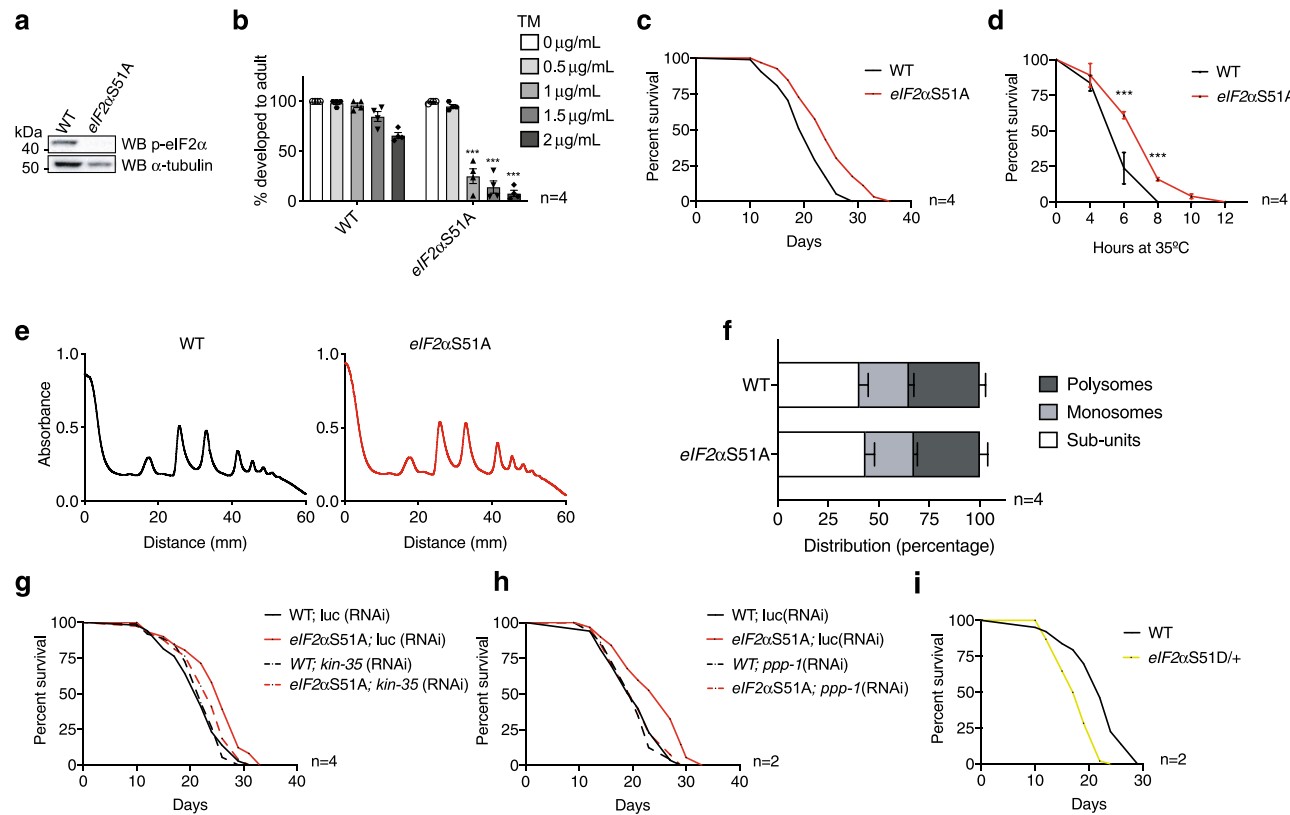

**Fig. 6 Direct ISR inhibition through phospho-defective *eIF2αS51A* mutations extends lifespan via eIF2B. a** Western blot of day 1 WT animals and *eIF2α*S51A mutants using anti-phospho-eIF2α (Ser51) and anti-α-tubulin antibodies. **b** Developmental tunicamycin (TM) resistance assay of WT animals and eIF2αS51A mutants treated with tunicamycin at the indicated concentrations (error bars represent means + SEM, two-way ANOVA Sidak's post hoc test, \*\*\*$p < 0.001$ vs. WT at respective tunicamycin concentration; $n = 4$ independent experiments with ≥20 animals each). **c** Survival of eIF2αS51A mutants compared to WT control animals (representative data from $n = 4$ independent experiments). **d** Thermotolerance assays of eIF2αS51A mutants compared to WT animals (error bars represent means ± SD, two-way ANOVA Sidak's post hoc test with \*\*\*$p < 0.001$ vs. WT controls; $n = 4$ independent experiments with 50 animals each). **e, f** Polysome profiling and quantification of day 1 WT worms and eIF2αS51A mutants (error bars represent means + SD, two-way ANOVA Dunnett's post hoc test; $n = 4$ independent experiments). **g** Survival of WT worms and eIF2αS51A mutants upon RNAi treatment targeting *kin-35* or control *luciferase* (representative data from $n = 4$ independent experiments). **h** Survival of WT animals and eIF2αS51A mutants upon RNAi treatment targeting *ppp-1* or control *luciferase* (representative data from $n = 2$ independent experiments). **i** Survival of heterozygous phospho-mimic *eIF2α*S51D/+ mutants compared to WT control animals (representative data from $n = 2$ independent experiments). See Supplementary Dataset 1 for survival statistics. See Supplementary Table 1 for statistics on thermotolerance assays. Source data are provided as a Source Data file.

the CRISPR/Cas9 system. Surprisingly, the homozygous *eIF2α*S51D mutation was lethal. To maintain the mutation at the heterozygous state, we used the genetic balancer ht2. Heterozygous *eIF2α*S51D/+ mutants displayed slow generation time compared to WT animals and reduced brood size (Supplementary Fig. 6f, g). In line with these data, they were short-lived (Fig. 6i). Together, these results demonstrate that constitutive ISR activation is detrimental for *C. elegans*.

Summarizing, we identified multiple longevity-associated eIF2 modulators in a genomic screen for longevity in *C. elegans*. ISR inhibition was linked to longevity and enhanced protein homeostasis, without a reduction of overall protein biosynthesis. Instead, selective translation of specific mRNAs was required for lifespan extension. Inhibiting the ISR using additional genetic or chemical interventions confirmed that fine-tuning of eIF2 leads to longevity.

## Discussion

Through a genomic mutagenesis screen for longevity in *C. elegans*, we found that distinct states of mRNA translation initiation and the ISR independently result in longevity. The unbiased screen revealed a mutation in *iftb-1*, the β subunit of the eIF2

complex, that decreased mRNA translation. Reduced translation is associated with longevity[29,31,37,38] and RNAi-mediated *iftb-1* knockdown inhibits protein synthesis, extending lifespan[25]. Thus, the genetic screening approach is validated for its capacity to identify longevity-associated pathways, in addition to changes in chemosensation and insulin signaling. More importantly, our data show that inhibition of the ISR extends lifespan, changing mRNA translation without reducing overall protein biosynthesis. This was achieved by genetically dominant mutations in eIF2Bγ/ *ppp-1*, as well as through the phosphorylation-deficient S51A mutation in eIF2α that fully impairs the ISR. While bulk protein synthesis remained unaffected in these mutants, translation of certain mRNAs was selectively altered, contributing to the longevity phenotype. In line with the lifespan extension, eIF2Bγ/ *ppp-1* and *eIF2α*S51A mutants displayed improved protein homeostasis, essential for cellular and organismal health.

Translation initiation and its modulation, including the ISR, have been deeply characterized in the general control pathway in yeast[20]. The class of general control non-derepressible (Gcn(−)) yeast mutants are unable to activate translation of the uORF-regulated transcription factor GCN4/ATF4 upon amino acid starvation[20]. In other words, Gcn(−) mutations attenuate the stress-induced expression of uORF-regulated genes such as

GCN4/ATF4, resulting in a state of ISR inhibition[20]. Mutations that reduce or abolish eIF2α phosphorylation, as in the partial *gcn-2* and *pek-1* loss-of-function and the *eIF2αS51A* mutants analyzed in this study, therefore belong to the Gcn(−) class. We also classified the dominant eIF2Bγ/*ppp-1* alleles as Gcn(−) mutations as they reduced uORF regulated *atf-4::GFP* expression under stress. eIF2B subunits have been identified carrying Gcn(−) mutations in yeast[39]. Upon eIF2α phosphorylation, eIF2 inhibits eIF2B[40] and mutations in eIF2Bβ/GCD7 and eIF2Bγ/GCD2 render eIF2B insensitive to its inactivation by phosphorylated eIF2[39,41]. These eIF2B variants are not inhibited despite an activated ISR. The eIF2Bγ/*ppp-1* mutants we found might have similar features regarding regulation by phosphorylated eIF2α and thus showed decreased ISR activity.

Translation initiation and the ISR are intimately linked. Our data suggest that a shift in the translatome, and not the loss of the ISR per se, was responsible for extending lifespan. Long-lived *daf-2*/insulin receptor mutants show changes in their translatome[42] and the extended lifespan of *daf-2; rsks-1*/S6K double mutants is mediated by the selective translational repression of the cytochrome *cyc-2.1*[43]. However, its translational efficiency was not changed in *ppp-1* mutants (Supplementary Dataset 2). Our study shows that Gcn(−) mutations change the translational efficiency of specific mRNAs that are required for the observed lifespan extension. This is in line with the regulation of aging at the level of mRNA translation. While it is not understood how *kin-35* mRNA is selectively recruited to polysomes, our data suggest that upregulation of KIN-35 contributes to a switch that enhances robustness. This is supported by the analysis of polysome-associated mRNAs in *ppp-1* mutants pointing to a broader change in the cellular dynamics of phosphorylation and dephosphorylation. Further biochemical and genetic analyses are needed to determine the downstream effects of KIN-35.

Previous data demonstrate that knockout mutations in *gcn-2* and *pek-1* do not affect WT *C. elegans* survival[44,45], which stands in an apparent contradiction with our data suggesting that single inhibitory amino acid substitutions in GCN-2 and PEK-1 actually extend lifespan. These discrepancies suggest that GCN-2 and PEK-1 kinases might have additional targets in addition to eIF2α. Alternatively, GCN-2 and PEK-1 might be part of larger protein complexes[46,47] that are affected by deletions or RNAi treatments but not by the more specific point mutations of the kinases.

A number of interventions that extend mouse lifespan show elevated ATF4 expression[48] and ATF4 is linked to lifespan extension via FGF21 in mice[49]. In addition, GCN4 is required in yeast to extend lifespan when translation is inhibited suggesting a beneficial effect of an activated ISR for longevity[50]. The relationship between lifespan extension and ISR activity has been further investigated in yeast as activated GCN2 extends lifespan in a GCN4/ATF4 dependent manner. In this model, the beneficial effect of ISR activation on replicative lifespan relies on the downstream GCN4/ATF4 effector that activates autophagy[51,52]. Further, pharmacological ISR activation is protective in a Huntingtin mouse model[53]. Nevertheless, deregulated activation of the ISR has also been correlated with cancer and diabetes[54,55]. The ISR is activated in neurogenerative disorders, traumatic brain injury, and Down syndrome[56–59]. Although the role of the ISR in longevity is thus unclear and is very likely to differ between cell types, no studies have yet formally tested how a direct modulation of the ISR affects mammalian survival. Our data show that reducing or fully abrogating the ISR in Gcn(−) mutants extended *C. elegans* lifespan. While the ISR is clearly required to cope with acute stress, the translatome changes in Gcn(−) mutants appear to support robustness and protein homeostasis. Further, our data do not argue against a possible lifespan extension through activation of ATF-4. While ATF4 likely enhances robustness in

conditions of reduced protein synthesis, selective mRNA translation downstream of eIF2B extends life through a fully independent mechanism that does not involve ISR activation or ATF-4 expression.

Pathological conditions associated with an increased ISR can be treated by reducing eIF2α phosphorylation or by interfering with the inhibition of eIF2B. Deletion of eIF2α kinases prevents pathology in a mouse model for Alzheimer's disease[56] and PKR knockout enhances cognitive function in a mouse model for Down syndrome[58]. This suggests a causal role of the ISR in these age-associated diseases. Further, memory is enhanced in mice heterozygous for the eIF2αS51A mutation[60]. Pharmacological inhibition of the ISR is possible using the small molecule ISRIB, which enhances memory, prevents neurodegeneration in prion disease, and reverses memory defects associated with traumatic brain injury[57,59,61]. Mechanistically, ISRIB stabilizes and activates eIF2B, which counters the effects of eIF2α phosphorylation[35,36]. In all, these data converge with the enhanced survival and robustness we observed in the Gcn(−) mutations in *C. elegans* eIF2Bγ/*ppp-1*. Our data show that tuning of eIF2 unexpectedly affected nematode survival as genetic or pharmacological inhibition of the ISR increased lifespan. This occurred without suppression of overall protein biosynthesis and might thus be a promising therapeutic approach to modulate the aging process.

## Methods

**C. elegans strains and culture.** All *C. elegans* strains were maintained at 20 °C on nematode growth medium (NGM) agar plates seeded with the *Escherichia coli* (*E. coli*) strain OP50 unless indicated otherwise[62]. To provide an isogenic background in all mutant strains, they were outcrossed against the wild type Bristol N2 strain. All strains used in this study are listed in Supplementary Table 2, including outcrossing information and source. Genotyping primers used in this study are listed in Supplementary Table 3. The strains *ppp-1(syb728)*, *ppp-1(syb691)*, Y37E3.10a *(syb1385)* (in the main text referred to as *eIF2αS51A*), and Y37E3.10a*(syb1567)* (in the main text referred to as *eIF2αS51D*) were generated by SunyBiotech (China) using CRISPR/Cas9; the correct sequence was verified by PCR and Sanger sequencing (Eurofins Genomics, Germany). The *eIF2αS51D* mutants are lethal in a homozygous state and hence maintained in a heterozygous state with the genetic balancer ht2. The balancer contains a pharyngeal GFP signal, which was used as a marker.

**Unbiased forward longevity screen.** The longevity screen was performed with the temperature-sensitive sterile strain CF512 *fer-15(b26); fem-1(hc17)*. L4 larvae were exposed to 0,3% ethyl methanesulfonate (EMS, Sigma) in M9 buffer for 4 h at room temperature. After recovery overnight, young P0 adult animals were transferred to new plates. Singled F1 progeny were allowed to lay eggs overnight. In the next generation, singled F2 progeny were allowed to lay eggs for 16 h. After egg-laying, F2 worms were stocked at 15 °C. F3 eggs were heat-shocked at 25 °C for 48 h to induce sterility and adult animals were scored twice a week for preliminary lifespan analysis. Mutants that outlived the non-mutagenized control by 20% (maximum lifespan) were selected for regular demographic lifespan analyses to confirm the longevity phenotype. After the lifespan assays, mutants with a mean lifespan extension above 18% compared to non-mutagenized CF512 controls were selected for whole-genome sequencing.

**Mutant mapping and sequence analysis.** Genomic DNA of selected long-lived strains was prepared using the QIAGEN Gentra Puregene Kit. Whole-genome sequencing was conducted on the Illumina HiSeq2000 platform. Paired-end 100 bp reads were used; the average coverage was larger than 16-fold. Sequencing outputs were analyzed using the CloudMap Unmapped Mutant Workflow pipeline on Galaxy[63]. The WS220/ce10 *C. elegans* assembly was used as the reference genome.

**Induction of endoplasmic reticulum stress with tunicamycin.** To induce endoplasmic reticulum (ER) stress with tunicamycin, worms were transferred on NGM plates containing different tunicamycin concentrations and 1% DMSO or control plates with 1% DMSO only. Standard treatment was at 10 μg/mL tunicamycin for 6 h at day 1 of adulthood, unless stated otherwise. Treatments for lifespan experiments, the compound screen, and developmental tunicamycin resistance assays are specifically described in the respective methods subsections.

**Induction of ER stress with dithiothreitol (DTT).** For the DTT treatment, an overnight culture of OP50 bacteria was 10-fold concentrated in an S-basal medium. Worms were transferred into 250 μL S-basal medium, 200 μL 10-fold concentrated

OP50 and 5 μL 1 M DTT (Sigma) diluted in S-basal. The volume was filled up to a total of 1 mL with S-basal (final DTT concentration: 5 mM). Worms were incubated for 2 h at 200 rpm.

**Lifespan assays.** Gravid day 1 adults were allowed to lay eggs for 5 h. The offspring was used for lifespan analysis. The L4 stage was defined as day 0 and more than 100 worms were used per strain and condition. Worms were kept at 20 °C on NGM plates seeded with OP50 E. coli at all times. The animals were transferred every second day to fresh plates until they reached the post-reproductive stage. Scoring was performed every second day by monitoring (touch-provoked) movement and pharyngeal pumping. Animals in RNAi lifespan assays were treated with RNAi from day 1 of adulthood and kept on NGM plates seeded with HT115 E. coli bacteria expressing control *luciferase* or targeting RNAi clones throughout the experiment. Animals in lifespan assays on estradiol valerate (Sigma), ISRIB (Sigma), or propafenone hydrochloride (Sigma) were transferred at the L4 stage to NGM plates containing 1% DMSO (Sigma) and 20 μM estradiol valerate/ISRIB/ propafenone hydrochloride or control plates with 1% DMSO only. Animals in lifespan assays on tunicamycin (TM, Sigma) were transferred on day 1 of adulthood to NGM plates containing 20 μg/mL TM and 1% DMSO or control plates with 1% DMSO only. Lifespan assays of heterozygous *ppp-1* animals were performed on F1 hermaphrodites after crossing of mutant hermaphrodites to WT males. Lifespan assays of heterozygous *eIF2αS51D/+* animals were performed using the genetic balancer ht2 (which can be recognized by expression of pharyngeal GFP). In all lifespan experiments, worms that had undergone internal hatching, vulval bursting, or worms crawling off the plates were censored. Throughout the experiment, strain and/or treatment were unknown to researchers. Data were assembled upon completion of the experiment. Statistical analyses were performed with the Mantel–Cox log-rank method in Prism (Version 8.2.0).

**Thermotolerance assays.** After an egg-lay, synchronized day 1 animals were transferred to 6 cm NGM plates containing OP50 and placed at 35 °C. Survival was scored for (touch-provoked) movement and pharyngeal pumping every 2 h until no survivors were left. Worms with internal hatching, vulval bursting, and worms crawling off the plates were censored. Throughout the experiment, strain and/or treatment were unknown to the researcher. Unless stated otherwise, at least three independent experiments were performed, error bars represent means ± SD and assays were analyzed by two-way ANOVA, Dunnett's, or Sidak's post hoc test as indicated.

**Dauer formation assays.** Gravid day 1 adults were allowed to lay eggs for 5 h at room temperature. For dauer formation assays, the offspring was shifted to 27 °C for 60 h. Dauer and non-dauer animals were scored according to their appearance (with at least 50 animals per strain). Dauer and non-dauer stages were verified by performing microscopy of the worm cuticle (Zeiss Imager Z1, Axio Cam ICC5, Zen 2.3 pro software). Images were analyzed with ImageJ.

**Motility assays.** Animals carrying the *unc-54P::Q35:YFP* (polyQ35) or the *unc54P::α-syn* transgene were grown on NGM plates seeded with OP50. For RNAi experiments, they were transferred at the L4 stage to plates seeded with HT115 bacteria expressing *luciferase* or candidate RNAi clones. On day 7 or 8 of adulthood, motility was tested by transferring single worms to M9, where they were allowed to acclimatize for 30 s, followed by the counting of body bends over 30 s. At least 12 worms were scored per experiment, genotype, and/or treatment. Throughout the experiment, strain, and/or treatment were unknown to the researcher. Unless stated otherwise, at least three independent experiments were performed, error bars represent means ± SD and assays were analyzed by one-way ANOVA, Dunnett's post hoc test.

**Western blotting.** For Western blotting, day 1 worms were collected in M9 and snap-frozen in liquid nitrogen. For protein extraction, worms were lysed in RIPA buffer (150 mM NaCl, 1% NP40, 0.5% sodium deoxycholate, 0.1% sodium dodecyl sulfate (SDS), 50 mM Tris-HCl, pH 8.0, completed with protease inhibitors), sonicated and spun down. Protein quantification was done by bicinchoninic acid assay (Pierce BCA Protein Assay Kit, Thermo Fisher). Equal amounts of protein were diluted in NuPAGE LDS Sample Buffer (4×, ThermoFisher) containing 50 mM Dithiothreitol (DTT). Proteins were then separated by reducing sodium dodecyl sulfate-polyacrylamide gel electrophoresis and transferred to nitrocellulose membranes (Amersham[TM] Hybond ECL), followed by blocking with milk or bovine serum albumin (BSA) and antibody labeling with specific antibodies to phospho-eIF2α (Ser51) (Cell Signaling; 1:2.000 in 2% BSA), puromycin (Merck Millipore; 1:10.000 in 5% milk), Living Colors GFP (Clontech; 1:5.000 in 5% milk), α-tubulin (Sigma; 1:10.000 in 5% milk) and histone H3 (Abcam; 1:10.000 in 5% milk). Immunolabeling was visualized using chemiluminescence kits (ECL, Amersham Bioscience) on a Chemidoc MP Imaging System (Biorad) and analyzed with the ImageLab Software (version 5.2, Biorad). Signals were quantified with ImageJ (version 1.51) and Prism (version 8.2.0). For Western blot analyses of compound-feeding experiments, worms were fed after hatching with 20 μM estradiol valerate (Sigma) and 1% DMSO, or 1% DMSO only. ER stress by DTT or tunicamycin was induced as described above. For Western blot

analysis at day 6 of adulthood (and corresponding day 1 control experiments), worms were transferred to NGM plates containing 10 μM 5-Fluoro-2′-deoxyuridine (FUDR, Sigma) at the L4 stadium. The collection of the Western blot samples was conducted simultaneously for day 1 and day 6 animals. Unless stated otherwise, at least 4 independent experiments were performed, error bars represent means ± SEM and assays were analyzed by one-way ANOVA, Tukey's, or Dunnett's post hoc test as indicated.

**SUnSET, puromycin incorporation.** To monitor protein synthesis in a non-radioactive manner using puromycin incorporation and puromycin detection based on Schmidt et al.[32], day 1 worms were collected in M9 and once washed into S-basal medium. For puromycin treatment, an overnight culture of OP50 bacteria was 10-fold concentrated in S-basal medium. Worms were then transferred into 250 μL S-basal medium, 200 μL 10-fold concentrated OP50, and 50 μL 10 mg/mL puromycin diluted in S-basal. The volume was filled up to a total of 1 mL with S-basal (final puromycin concentration: 0.5 mg/mL). Worms were incubated for 3 h at 200 rpm. Afterwards, they were washed 3 times in S-basal and snap-frozen in liquid nitrogen. Worms were kept on ice after the puromycin treatment. Protein extraction and Western blot using an anti-puromycin antibody (Merck Millipore) were performed as described before.

**35S-methionine labeling.** To monitor translation rates, 35S-methionine labeling was performed based on Hansen et al.[25]. OP50 bacteria were cultured overnight in LB medium (1 mL/sample) containing 15 μCi of 35S-methionine and concentrated 10-fold. Synchronized day 1 worms were added to the mix and incubated for 3 h at room temperature. Worms were washed twice with S-basal and incubated in nonradioactive OP50 (10-fold concentrated). Worms were then washed twice with S-basal medium before three freeze/thaw cycles using liquid nitrogen. Worm pellets were boiled in 100 μL 1% SDS and centrifuged 2 min at 2000g to remove cuticles. Supernatants were submitted to trichloroacetic acid precipitation. Protein pellets were neutralized with 20 μL of 0.2 M NaOH. Proteins were solubilized with 180 μL of 8 M urea; 4% chaps; 1% DTT. Protein concentrations were measured using Bradford reagent and 35S radioactivity was measured by liquid scintillation. Unless stated otherwise, at least five independent experiments were performed, error bars represent means ± SEM and assays were analyzed by one-way ANOVA, Dunnett's post hoc test.

**Polysome profiling.** For the analysis of translation via polysome profiling based on Ding and Großhans[64], synchronized gravid day 1 adults were grown on NGM plates seeded with OP50. Per genotype and replicate, ~12,000 worms were harvested and washed twice with M9, once with M9 supplemented with 1 mM cycloheximide (Sigma) and once with lysis buffer (20 mM Tris pH 8.5, 140 mM KCl, 1.5 mM MgCl2, 0.5% Nonidet P40, 1 mM DTT, 1 mM cycloheximide). Worms were pelleted and resuspended in 350 μL cold lysis buffer supplemented with 1% sodium deoxycholate (DOC, Sigma). Resuspended worms were lysed using a chilled Dounce homogenizer. Ribonuclease inhibitor RNasin (Promega) was added to samples used for RNA sequencing or quantitative PCR (qPCR) at a concentration of 0.4 U/μL. Samples were then mixed and incubated on ice for 30 min, followed by a centrifugation step (12,000g, 10 min, 4 °C) for clearance. The pellet was discarded and the RNA concentration of the supernatant was estimated by absorbance measurement at 260 nm.

To prepare sucrose gradients, 15% (w/v) and 60% (w/v) sucrose solutions were prepared in basic lysis buffer (20 mM Tris pH 8.5, 140 mM KCl, 1.5 mM MgCl2, 1 mM DTT, 1 mM cycloheximide). Linear sucrose gradients were produced using a Gradient Master (Biocomp). Equivalent amounts of sample (around 400 μg RNA) were loaded on the gradient and centrifuged at 39,000g for 3 h at 4 °C, using an Optima L-100 XP Ultracentrifuge (Beckman Coulter) and the SW41Ti rotor. To analyze the sample on the gradient during fractionation, absorbance at 254 nm was measured and recorded (Econo UV monitor EM-1, Biorad) using the Gradient Profiler software (version 2.07). Gradient fractionation was performed from the top down using a Piston Gradient Fractionator (Biocomp) and a fraction collector (Model 2110, Biorad). Gradients were fractionated into 20 fractions of equal volume. In an initial experiment, the ribosomal fractions were validated by analyzing RNA from each fraction via agarose gel electrophoresis. The 18S and 28S rRNA signals were used as indicators for the 40S ribosomal subunit, the 60S ribosomal subunit and fully assembled ribosomes. Quantification of the ribosomal complexes was performed using Image J and statistically analyzed with Prism. Unless stated otherwise, at least four independent experiments were performed, error bars represent means ± SD and assays were analyzed by two-way ANOVA, Dunnett's post hoc test.

For more precise analysis of ribosomal fractions, they were collected by hand according to their absorbance profile; for RNAseq and qPCR analyses, one fraction for 80S ribosomes and one for polysomes (excluding disomes) was collected per sample. RNA extraction from total lysates and from each fraction was performed using the Direct-zol RNA MicroPrep Kit (Zymo Research) according to the manufacturer's recommendations.

**Polysome sequencing.** For polysome sequencing, monosome extracts, polysome extracts (without disomes), and corresponding total RNA were collected as detailed

above. cDNA libraries were generated with ribosomal RNA depletion at the Cologne Center for Genomics and sequenced on the Illumina HiSeq2000 platform.

For data analysis, raw reads from all RNAseq and polysome sequencing replicates were mapped to the *C. elegans* reference genome (ENSEMBL 91) using HISAT2 (v2.1.0)[65]. After guided transcriptome assembly with StringTie (v1.3.4d), transcriptomes were merged with Cuffmerge, and quantification was performed with Cuffquant[66]. The analysis for differential gene expression for total, monosomal, and polysomal RNA was performed with Cuffdiff (Cufflinks v2.2.1)[67,68]. To analyze the translatome, the abundance of each mRNA in the polysomal fraction was normalized to its abundance in the total input mRNA. Respective normalized values were used to identify changes between different conditions using the Student's *t* test. For further analyses, we only included the mRNAs that were found significantly changed in both *ppp-1* mutants. For each mRNA, the mean *p* values and the mean log-2 fold change of both *ppp-1* mutants were used. David analysis was performed to identify significantly enriched gene ontology terms[69].

**RNAi experiments**. For RNAi-mediated knockdown of specific genes, HT115 bacteria carrying vectors for dsRNA of the target gene under a promotor inducible by isopropyl β-D-1-thiogalactopyranoside (IPTG) and ampicillin resistance were used. Bacteria were seeded on NGM plates containing 100 μg/μL ampicillin (Merck Millipore) and 1 mM IPTG (Roth). After egg-lay, worms were grown on regular NGM plates seeded with OP50 bacteria until the L4 stage and then transferred to RNAi plates. RNAi against *luciferase* was used as nontargeting control. All RNAi clones were obtained from the Ahringer and Vidal RNAi libraries[70,71]. Clones were validated by plasmid purification (QIAprep Spin Miniprep Kit, Qiagen) and sequenced using the L4440 seq RV primer.

**Selective RNAi screen for suppressors of *ppp-1* motility**. Synchronized worms of the *ppp-1(wrm10)* strain crossed to *mLs133[unc-54P::Q35:YFP]* animals (polyQ35; *ppp-1(wrm10)*) and control *mLs133[unc-54P::Q35:YFP]* worms (polyQ35 WT) were grown to the L4 larval stadium. Animals were then placed on NGM plates containing 10 μM FUDR to inhibit the development of progeny. Plates were seeded with HT115 bacteria expressing selected RNAi clones to knock down specific genes in the nematodes. At day 8 of adulthood, the motility of polyQ35; *ppp-1(wrm10)* as well as polyQ35 WT worms was assessed on *luciferase* control RNAi and 66 RNAi treatments targeting mRNAs enriched in *ppp-1* polysomes. To test motility, 15 worms were picked into the center of a 10 mm circle on an unseeded NGM plate and their ability to leave the circle after one minute was scored. For more reliability, four experiments were performed for the control conditions (polyQ35 WT and polyQ35; *ppp-1(wrm10)* on *luciferase* RNAi; error bars represent means ± SD).

RNAi treatments rescuing the polyQ35; *ppp-1(wrm10)* motility phenotype to at least 50% compared to the polyQ35; *ppp-1(wrm10)* control on *luciferase* RNAi were validated by full motility assays (without the usage of FUDR) counting body bends over 30 s in liquid. In a counter screen, the effect of the RNAi treatments on polyQ35 WT animals was tested. To this end, young worms were treated as described before and the motility on day 6 of adulthood was scored. If motility of polyQ35 WT worms treated with RNAi against candidate mRNAs was significantly lower compared to animals treated with *luciferase* RNAi, candidates were excluded from further analysis.

**Worm imaging**. For worm imaging, animals were arranged in stacks on unseeded NGM plates and kept on ice. Images were taken with a fluorescence microscope (Leica M165FC) and a camera (Leica DFC 3000G). Images were aquired and analyzed with the Leica Application Suite X (Version 3.4.1.17822). Images were quantified with ImageJ (Version 1.51). Scale bar is indicated in the figure legends.

**Compound screen**. To identify compounds inhibiting the ISR, synchronized *atf-4P::GFP::unc-54* 3'UTR L4 animals were transferred to NGM plates without or with 4 μg/mL tunicamycin. Plates were supplemented with 1% DMSO (Sigma) as control, or with 1% DMSO and 20 μM estradiol valerate (Sigma), ISRIB (Sigma), GSK2606414 (Calbiochem), propafenone hydrochloride (Sigma), azadirachtin (Sigma) or estriol (Sigma), respectively. Day 1 animals were analyzed by fluorescence microscopy as described above.

**Developmental tunicamycin resistance assays**. For developmental tunicamycin resistance assays, NGM plates supplemented with 10 μg/mL tunicamycin and control plates without tunicamycin were used (seeded with OP50 bacteria). 50–80 synchronized eggs per genotype and/or condition were added to the plates. Development to the adult stage was scored after 4 or 5 days. Unless stated otherwise, at least four independent experiments were performed, error bars represent means ± SEM and assays were analyzed by two-way ANOVA, Sidak's post hoc test.

**Pharyngeal pumping**. Pharyngeal pumping rates of synchronized animals were measured at day 1 of adulthood by counting pharyngeal contractions per worm during 30 s. Per experiment and genotype, at least 15 worms were analyzed.

Throughout the experiment, strain and/or treatment were unknown to the researcher. Error bars represent means ± SD.

**Generation time**. For generation time assays, synchronized eggs were allowed to develop into adult worms on single plates until they laid the first egg, which was defined as the generation time. After 55 h, animals were scored every hour with 15 worms being analyzed per experiment and genotype. Throughout the experiment, strain and/or treatment were unknown to the researcher. Error bars represent means ± SD.

**Brood size assays**. For brood size assays, synchronized L4 worms were placed on individual NGM plates seeded with OP50 bacteria. Worms were transferred to fresh plates every 24 h until no more eggs were laid. The number of viable progeny on each plate was counted and summed up for each individual parental worm. Per experiment, genotype and/or condition, at least 15 parental worms were analyzed. Error bars represent means ± SD.

**qRT-PCR (qPCR)**. For qPCR analyses, day 1 worm samples or indicated samples from ribosome profiling were collected in TRI Reagent (Zymo) and frozen in liquid nitrogen. RNA extraction was performed using the Direct-zol RNA MicroPrep Kit (Zymo Research) according to the manufacturer´s recommendations, followed by cDNA synthesis (iScript cDNA Synthesis Kit, BioRad). qPCRs were performed using Power SYBR Green PCR Master Mix (Applied Biosystems) on a ViiA 7 Real-Time PCR System (Applied Biosystems). Expression levels of the gene *act-1* were used as internal control for normalization. All qPCR primer sequences can be found in Supplementary Table 4. Unless stated otherwise, at least three independent experiments were performed, error bars represent means ± SEM and assays were analyzed by two-way ANOVA, Tukey's post hoc test.

**Statistical analysis**. Results are presented as means ± SD or means ± SEM. Statistical tests were performed using one-way or two-way ANOVA with Sidak's, Dunnet's or Tukey's multiple comparison test. Significance levels are depicted in the figures and specified in the figure legends. Experiments were carried out with at least three biological replicates unless noted otherwise.

**Reporting summary**. Further information on research design is available in the Nature Research Reporting Summary linked to this article.

## Data availability
The RNA sequencing data in this publication have been deposited in NCBI's Gene Expression Omnibus and are accessible through GEO Series accession number GSE144607. All other data are available in the main text or the Supplementary Materials. The source data underlying Figs. 1j, 1k, 2a–e, 4d, 5c, 6a, b, e, f, and Supplementary Figs. 1c, 2a–c, 3b–e, 4b, 4e–h, 5a, 6a–c, and 6e–g are provided as a Source Data file. Source data are provided with this paper.

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

## Acknowledgements

We thank all Denzel laboratory members for helpful discussions throughout this project. We thank the Caenorhabditis Genetics Center (CGC) and T. Keith Blackwell for worm strains. The *atf-4P::GFP::unc-54 3′UTR* reporter was generated in David Ron's laboratory by Chi Yun and Cole Haynes. We thank Franziska Metge, Sven Templer, and Jorge Boucas as well as all members of the bioinformatics core facility at MPI AGE. We thank the Cologne Center for Genomics for sequencing. We further thank William B. Mair, David Ron, Peter Walter, Dario R. Valenzano, Matias D. Hartman, and Kira Allmeroth for valuable comments on the project and the paper. *Funding*: L.E.W. was supported by the Cologne Graduate School of Ageing Research. This work was supported by the European Commission (ERC-2014-StG-640254-MetAGEn). Figure 3a and Supplementary Fig. 3a were created with BioRender.com

## Author contributions

M.J.D., L.E.W., and M.S.D. conceived the study. All experiments were performed by M.J.D., L.E.W., and R.B. The paper was written and edited by M.J.D., L.E.W., and M.S.D.

## Funding

## Competing interests

The authors declare no competing interests.
