## [Peer Review File · Nature Communications]

Reviewer comments, first round:

Reviewer #1 (Remarks to the Author):

In their manuscript "eIF2B extends lifespan through inhibition of the integrated stress response" M. Derisbourg and co-authors establish the connection between integrated stress response (ISR) modulation and longevity in *C. elegans*. To investigate novel alleles that are linked to increased lifespan, the authors of this study performed a forward genetic screen for long-lived mutants and identified several hits in genes that were already known to regulate the ISR (*ppp-1*, *gcn-2*, *pek-1*). After confirming the longevity phenotypes of the identified alleles in outcrossed and CRISPR/Cas9-generated worms, the authors additionally showed the involvement of the same alleles in proteotoxic resilience (heat stress, resistance against protein aggregation). The authors put the newly acquired data in the multicellular organism *C. elegans* into context with the *Gcn(-)* phenotype, which has already been studied in yeast. Moreover, the authors applied mid-life pharmacological inhibition of the ISR, which caused similar longevity effects like the ISR mutants *ppp-1*, *gcn-2*, and *pek-1*. Importantly, the authors ruled out that *Gcn(-)* longevity is not dependent on the attenuation of global translation but is rather dependent on the upregulation of specific mRNA transcripts. A following transcriptome approach identified several mRNA transcripts that are differentially translated in hypoactive ISR worms. Further genetic experiments conducted by the authors resulted in the finding that the predicted kinase *kin-35* is fully responsible for the longevity phenotype of hypoactive ISR mutants (*ppp-1*), implying a fundamental role of *kin-35* in cellular processes that regulate longevity through the modulation of ISR.

While the link between ISR and longevity/proteostasis has been shown before in yeast and other model systems, the described findings might be of relevance to further understand the physiological role of the ISR in the aging process. The most interesting part of the data addresses how ISR inhibition affects longevity, however, it remains incomplete. The authors should address the role of the identified kinase *KIN-35* in more detail to strengthen the mechanistic novelty of their findings.

Major points and concerns:

Interestingly the authors suggest that the lifespan extension caused by ISR inhibition is not mediated by overall changes in protein translation rate. Instead, they identified enhanced translation of certain mRNAs including *kin-35*. Besides *kin-35(RNAi)* mediated suppression of the extended lifespan phenotype of *ppp-1* mutants it would be interesting to test the role of the kinase activity of *KIN-35*. Therefore, the authors should use kinase inactive *kin-35* mutants to address genetic suppression of double mutants. If the kinase activity is required it would be interesting to test direct regulation of the ISR biochemically. Moreover, it would be interesting to test if *KIN-35* overexpression shows lifespan extension.

F2F: The dynamic range of the tunicamycin developmental assay is low and scarcely allows the claimed statement that *eIF2aS51A* mutants are hypersensitivity compared to WT worms without a statistical evaluation or stated p-value. Increasing the dynamic range of the assay might be achieved with lower tunicamycin concentrations (0.1 µg/ml, 0.5 µg/ml, 1 µg/ml). The negative control should be conducted with the solvent in which tunicamycin is diluted (e.g. DMSO). In case DMSO was not used as the solvent, a clarification in the material and methods about the solvent should be added.

F2G: The hypoactive ISR strain *eIF2aS51A* is long-lived under non-stress conditions. Following the narrative of the manuscript (discussion), the lifespan should be drastically reduced in *eIF2aS51A* worms under conditions where ISR activation is crucial for reestablishing cellular homeostasis (amino acid starvation, ER-stress e.g. tunicamycin treatment, viral infection e.g. orsay virus infection). This information would contribute to a clearer and better understanding whether the *eIF2aS51A* strain behaves according to the narrative of the manuscript and whether chronic ISR

ablation is detrimental under stress conditions.

Minor points and concerns:

Non-breaking hyphens should be used for allele names.

For better transparency about the quality of data, it is necessary to show each data point in the bar plots instead of the average + SEM. The visualization of each datapoint across independent experiments allows the reader to validate the robustness of the underlying biology. The data visualization like in Fig1I should be adapted throughout the whole manuscript for the bar plots.

F2C: Quantification of GFP/TUB values seem not proportional to the shown western blot and the fluorescent microscopy pictures in Extended Data F2B.

F2D: The blot could have a slightly higher exposure for better visibility. The representative blot is not reflecting clearly the quantification. The Y-axis labeling [%] of the quantification is missing. Total eIF2A (unphosphorylated) levels are not shown throughout the whole manuscript. In case that there is no available antibody for total eIF2A in the worm, the genetic data in the eIF2aS51A strain is slightly compensating for this drawback, making it a minor instead of a major concern.

F2I: The values of the peIF2a/TUB quantification of seem not proportional to the shown western blot.

F3B: The negative control rsk-1(sv31) shows a robust decrease in overall puromycin incorporation and therefore global cellular translation. It would be highly desirable to repeat the experiment ones more to increase the number of independent experiments from n=2 to n=3. Especially since this positive control beautifully shows the efficacy of this assay.

F3C & F3E: An overlay of the polysome profiling data should increase comparability between the strains (colors, transparent lines). In case that all three traces share exactly the same flow, the chosen representation might be better.

Extended Data F3: Are total eIF2a levels regulated in D6 worms? Is there a difference between WT and ppp-1 strains in total eIF2a levels?

Fig4: The elimination process of the other target mRNAs after Fig4E is non-transparent. In case that the authors tested the remaining 5 targets for their longevity effect, an additional statement and the extended data of the lifespans would be appreciated.

Fig4F & G: legend is overlapping between different genotypes, which is misleading

Reviewer #2 (Remarks to the Author):

In this manuscript the authors studied the link between Eif2B/Eif2 kinases and longevity in *C. elegans*. The authors show data where eIF2B extend life span through ISR inhibition and independently of global protein synthesis. Alternatively, the authors suggest that it is dependent of the Kin35 kinase which is preferentially translated under these conditions.

In this manuscript the authors provide a solid dataset of genes that may be interesting to investigate during aging in future studies. Despite that the involvement of the eIF2 pathway in aging was nicely shown in other species (Ref 16-novelty??). Moreover the authors fall short to show some mechanistic directions. The authors do suggest the involvement of Kin35 but not really show mechanistically how this is done and what is the role of Kin35 in longevity. Moreover they didn't show how Kin35 is translationally enhanced. These are major concerns that prevent me to recommend this manuscript for publication in Nature comms.

Other comments

1. Fig1, they investigated 101 genes of 127, what is the fate of the rest?
2. Figure 1h is shown at day 1 when eif2 is apparently not induced in WT, what happens if they repeat the experiment in day 6 where eif2a is active and is induced compared to day 1 as in extended figure 3a.
3. Everywhere where the author show eIF2a phosphorylation this should be normalized to total eIF2a/tubulin instead of Tubulin alone as shown in several places throughout the manuscript.
4. Figure 2a the dashed lines and the colors are hard to follow.
5. In Fig1 and Fig2 the WBs are done with DTT while the survival assay is done with Tm, lack of consistency. What is the status of eIF2a-p under Tm ?
6. The concentrations of Tm are very high and kill most of the WT more than 70% with the lowest concentration that was used. . Tm toxicity can be unspecific, I wonder what happens if they repeat this with lower concentrations of Tm that are enough to phosphorylate eIF2a and then to compare % of adult.
7. What is the status of the other UPR arms in these mutants? Is there a compensation of the other pathways that cause miRNA degradation that in turn stabilize Kin35 and enhance translation?
8. In C elegans most of the UPR signaling was shown to occur through IRE1 whereas PERK and ATF6 pathways were modestly involved (initially reported in PLoS Genet. 2005 Sep;1(3):e37), how do the authors reconcile their data with these results especially in the context of UPR induction?
9. Instead of blocking eif2a phosphorylation it would also be of interest to evaluate how maintaining its phosphorylation through the use of compounds such as Sephin1, Salubrial or guanabenz (the last 2 having already been used in C elegans) would affect the observed phenotypes (aging/longevity)
10. Title is misleading, how eIF2B does extend lifespan? Is it its activity that is important? not clear from the title.

Reviewer #3 (Remarks to the Author):

In this manuscript, Derisbourg and colleagues investigate the role of integrated stress response (ISR) in regulation of ageing. A core event of ISR is the phosphorylation of eukaryotic initiation factor 2a (eIF2 α) by one of four eIF2 α kinases, i.e. PERK, PKR, HRI and general GCN2. In an effort to identify novel longevity modulators, the authors performed an unbiased forward screen by using 0.3% ethyl methanesulfonate (EMS) in a conditionally sterile C. elegans strain. The screen revealed that mutations in the *ppp-1/eIF2B γ* , *pek-1/PERK* and *gcn-2/GCN-2* genes were able to extend lifespan and confer protection against proteotoxic stress caused by heat shock and the accumulation of aggregation prone proteins such as polyQ35. These data suggest a causative relationship between ISR and longevity. Moreover, an engineered phospho-defective eIF2A Δ S51A mutant was found to be long-lived and heat resistant. Consistently, pharmacological inhibition of ISR by estradiol valerate reduced GFP induction that occurs in animals expressing the *atf-5P::GFP* reporter after tunicamycin treatment. By contrast, ISR induction with propafenone hydrochloride further increased GFP expression in this reporter strain and shortened the lifespan of wild-type animals. *ppp-1* mutants exhibited comparative protein synthesis rates as the wild-type controls as evidenced by measurements of incorporated radioactive methionine and surface sensing of translation (SUnSET). In addition, polysome profiling failed to detect any difference in the overall distribution and abundance of wild-type and eIF2A Δ S51A mutant animals. Together, these data suggest that the extended lifespan of *ppp-1* mutants is uncoupled from reduced global protein synthesis. Further analysis of polysome-associated mRNAs showed that genes encoding proteins involved in phosphorylation events are enriched in *ppp-1* polysome fractions compared to wild-type controls. Interestingly, translational efficiency of specific mRNAs, including *kin-35* was increased in *ppp-1* mutants. Knockdown of *kin-35* suppresses the enhanced longevity of *ppp-1* and eIF2A Δ S51A mutant animals and abolished the motility of polyQ35; *ppp-1(wrm10)* animals. These findings suggest that *kin-35* is required for lifespan extension and increased proteostasis in animals carrying the above mentioned *Gcn(-)* class mutations. This manuscript presents interesting findings which suggest a role for ISR in regulation of ageing. However, there are several experimental and interpretation issues that need to be further addressed.

Major concerns

The title of the manuscript pertains to the mechanism by which eIF2B enhances longevity. This mechanism seems to rely on the inhibition of ISR. The authors have attempted to inhibit ISR in a pharmacological manner by using estradiol valerate. This compound (as every compound used in a living organism) can have pleiotropic effects, especially if it is a steroid hormone with the potential to activate nuclear hormone receptors (there are numerous in *C. elegans*). There are other available chemical compounds in the literature that can inhibit ISR specifically. The most prominent example is ISRIB which enhances the assembly of the eIF2B complex (doi:10.1111/febs.15073). ISRIB is a drug-like small molecule that suppresses ISR (Costa-Mattioli and Walter, 2020, *Science*, 368, 384) and binds to eIF2B leading to its activation (Zyryanova et al, 2018, *Science*, 369, 1533-1536). Two very interesting studies published recently provided intriguing findings about an OMA-1-DELE-1-HRI axis which relays mitochondrial stress to the ISR (Guo et al, *Nature* 2020 and Fessler et al, *Nature* 2020). The authors have exclusively examined ISR activation upon administration of DTT, an ER stressors. It would be nice if they could extend some of their observation to mitochondrial stress (such as CCCP, oligomycin or antimycin).

The existing literature suggests that mutations in neither PEK-1/PERK nor GCN-2 extend *C. elegans* lifespan (Henis-Korenblit et al, *PNAS* 2010 and Baker et al, *Plos Gen* 2012). The second study specifically suggests that GCN-2 is only required for the extended longevity of mitochondrial mutants (like *isp-1* or *clk-1*). Since the strains used in this study originated from random mutagenesis (with the possibility of accumulation of background mutations) this discrepancy should be convincingly explained. Does genetic inhibition of these three components via RNAi also extend lifespan of wild type animals? What is the nature of the described mutations? Are they null or hypomorphs?

Do phosphomimetic mutants (ex. eIF2aS51D or eIF2aS51E) show declined protein homeostasis and lifespan? Is any phosphatase [ex. Type 1 protein phosphatase (PP1) (or other)] able to compete/dephosphorylate eIF2? In this case does PP1 regulate protein homeostasis and lifespan?

Regulation of eIF2 GTPase by typical circle interventions: Can the above results in protein homeostasis and lifespan be confirmed by constitutively active and dominant negative eIF2a. What's the role of eIF2 GTPase-activating protein eIF5?

Given that activation of ISR has been associated with several pathological conditions, such as neurodegenerative diseases, diabetes and metabolic disorders, among others, it would be of interest to investigate how manipulation of ISR in specific cell types, for example in specific nematode neurons, can influence whole organism survival.

The fact that heterozygous *ppp-1* mutants are long-lived suggest that the *ppp-1(wrm10)* and *ppp-1(wrm15)* mutations are genetically dominant. However, it is not clear to me how the finding that *ppp-1* RNAi abolishes longevity of *ppp-1* mutant animals supports the abovementioned conclusion (please, see lines 90-92, page 4).

The enhanced resistance of *ppp-1* mutants to heat stress might be due to the activation of heme-regulated eIF2a kinase (HRI) that reportedly involves the HSP90 and HSP70 proteins (Lu et al, 2001, *Mol Cell Biol* 21: 7971 – 7980). Did the authors check this possibility?

The authors claim that pharmacological inhibition of ISR by estradiol valerate reduces the expression of an *atp-5P::GFP* reporter and extends *C. elegans* lifespan. By contrast, ISR induction with propafenone hydrochloride further elevated *atp-5P::GFP* expression and shortens lifespan. However, both compounds i.e. estradiol valerate and propafenone hydrochloride have been reported to act as UPRER suppressors (Halliday et al, 2017, *Brain: a journal of neurology*, 140, 1768-1783). Hence, their differential effects on *atp-5P::GFP* needs further investigation.

The authors claim that estradiol valerate suppressed eIF2A phosphorylation upon DTT treatment. However, the decrease observed is not statistically significant compared to vehicle (DMSO)-treated animals exposed to DDT as Fig. 2i showed.

Although ATF4 is the master transcription factor in ISR, it acts in combination with other proteins

and transcription factors in certain cases and also it can be regulated at the transcriptional level by TFEB and TFE3 transcription factors during ISR. The authors need to provide further mechanistic insight into how modulation of ISR influences lifespan.

Among the seven genes that suppress the motility of polyQ35; ppp-1(wrm10) animals when knocked down, the authors chose to investigate further the effects of only two genes, C01A2.5 and M04F3.3. It appears, therefore, that the selection of genes to further examine and also the interpretation of the results is biased. For example, the D1014.3 gene encodes a protein that is reportedly involved in IRE-1-mediated unfolded protein response. This gene as well as the other ones that have been shown to be non-toxic to wild-type animals should also be further tested for their effects on longevity and the maintenance of proteostasis in ppp-1 mutant animals.

More importantly, there are proteins that show very high amino acid similarity to the so-called KIN-35 protein based on blast analysis. The authors should examine whether selective translation of the corresponding mRNAs (for example, W09C3.1, T11F8.4) is also required for lifespan extension and increased homeostasis in Gcn(-) mutants.

Authors show that KIN-35 is essential for inhibited ISR-mediated lifespan extension. To see if KIN-35 can sufficiently enhance longevity they must show that KIN-35 overexpression in wild type background can increase lifespan. Also, tissue-specific overexpression of KIN-35 can shed light on the mechanism through which this protein affects lifespan. For example, does KIN-35 affect lifespan through systemic or cell autonomous effects? Also, more experiments are required to prove that KIN-35 improves proteostasis, such as analysis of reporter systems known to be affected by proteostatic factors.

Several reports show that different levels of ISR alterations can have different effects on homeostasis and survival (Pakos-Zebrucka, 2016, EMBO reports). Excessive increase of ISR can lead to cell death, while mild increase improves survival. The authors must check to what extent ISR levels were affected in their experiments, e.g. through biochemical analysis of proteostatic markers.

In animal models it has been shown that, when activity of genes regulating protein synthesis is altered, feedback mechanisms affect insulin signaling, hence longevity. To clarify this, authors must measure activity of insulin signaling in the whole body of the animals, or in a cell autonomous way.

It has been reported that, when ISR is at a very high level, ISR inhibitors are insufficient to reduce ISR-induced cytoprotection (Costa-Mattioli et al., 2020). The authors must show that the results of ISR-inhibitors treatments are independent of the stress levels of the worms prior to their treatment.

Minor comments

More total protein measurement methods (such as western blot analysis with more than one control antibodies and fluorescent reporters) must be applied to further indicate that total proteins levels are not affected by reduced ISR.

The authors observed increased motility of polyQ35 expressing nematodes upon ISR inhibition. Are the polyQ35 aggregates less in ppp-1(wrm10), ppp-1(wrm15), gcn-2(wrm4) and pek-1(wrm7) mutants? Is this a direct effect of general protein translation modulation or other cytoprotective mechanisms (e.g. autophagy or proteasome) are induced upon ISR inhibition?

Is ISR inhibition also protective against other models of proteostasis collapse, such as models of Alzheimer's disease (Tau-, A β - overexpressing nematodes) and Parkinson's disease (a-synuclein overexpressing nematodes)?

Do pharmacological ISR inhibition and/or ppp-1(wrm10), ppp-1(wrm15), gcn-2(wrm4) and pek-1(wrm7) mutants display improved cognitive and muscle function during ageing?

What do the dashed lines in Fig. 2a and Extended Data Fig. 2a denote? I suppose they correspond

to survival curves of ppp-1(wrm10) and ppp-1(wrm15) animals subjected to ppp-1 RNAi. The lifespan curves are not easily discriminated in Figures 2a, 4f & 4g.

The authors could check the expression of additional genes and not just atf-5 (i.e. CHOP, ATF4 and/or GADD34) known to be induced upon ISR.

As the authors state, they use the RNAi clones from the available libraries (Ahringer and Vidal). It is crucial that at least for those presented in the figures they provide a RT-PCR quantifying the reduction of mRNA level upon RNAi treatment.

Asterisks are missing in Extended Figure 4a.

Correct legend in Figure 4e. WT condition is not shown in the image as described in the legend. Or maybe the Figure title is wrong, If the graphs in the figure are title correct, then why the two different mutants are differently affected by some of the genes?

Line 73: Replace "Fig. 1b" with "Fig. 1c".

Lines 110-113 and Fig. 2f: Is the effect of 2 or 4 µg/mL TM statistically different between WT and eIF2aS51A mutants to support hypersensitivity to ER stress?

Line 121 and Extended Data Fig. 2g: Control chemical treatments without TM are missing in order to assess uORF-regulated translational activation via atf-5 reporter.

Fig. 3a and b: Why were different positive controls used in the radioactive methionine labeling (a) and SUnSET (b) (ifg-1 and rsk-1/S6K respectively)?

Lines 165-167 and Extended Data fig. 4a: C01A2.5 seems to have motility reduction in WT animals [on top of "shortened WT lifespan suggesting general toxicity" (lines 173-175 and Extended Data fig. 4c)]. Was that the reason at first place that C01A2.5 RNAi wasn't included (or presented) in lifespan analyses of ppp-1 and eIF2a-S51A longevity (Fig. 4f-g)? It would make sense to mention/discuss it before omitting one of the two candidates at the immediate following experiment.

The Authors might consider moving Fig. 3 to extended data and move some extended data to main figures (ex. Extended Data Fig. 2a, b, f, g, h).

The meaning of ISR induction or inhibition throughout the ms is sometimes confusing to the reader. It is used both when referring to eIF2a/eIF2b mutations (as these two genes are critical to the ISR cascade) and to the activation/inhibition of the downstream transcription factors (GCN4 homologues) (compare lines 179 and 182). It would be clearer if the IRS inhibition/induction would refer only to the end effect (ATF-5 protein levels).

The authors should discuss other studies' findings, showing that increased ISR extends lifespan in yeast (Postnikoff et al., 2017, Microbe cell).

REVIEWER COMMENTS

*We would like to thank all reviewers for their insightful comments. We have now generated significant new data that are included in an updated manuscript in which we also changed the paper's structure to better highlight the important conceptual advances of the study. Motivated by a recent manuscript by the Ewald and Blackwell labs (doi: 2020.11.02.364703), we have also decided to re-label the worm gene T04C10.4 that was previously called *atf-5* and are now calling it *atf-4*. This is also more consistent with the mammalian nomenclature. Below we will address the reviewers' comments point by point.*

Reviewer #1 (Remarks to the Author):

In their manuscript "eIF2B extends lifespan through inhibition of the integrated stress response" M. Derisbourg and co-authors establish the connection between integrated stress response (ISR) modulation and longevity in *C. elegans*. To investigate novel alleles that are linked to increased lifespan, the authors of this study performed a forward genetic screen for long-lived mutants and identified several hits in genes that were already known to regulate the ISR (*ppp-1*, *gcn-2*, *pek-1*). After confirming the longevity phenotypes of the identified alleles in outcrossed and CRISPR/Cas9-generated worms, the authors additionally showed the involvement of the same alleles in proteotoxic resilience (heat stress, resistance against protein aggregation). The authors put the newly acquired data in the multicellular organism *C. elegans* into context with the *Gcn(-)* phenotype, which has already been studied in yeast. Moreover, the authors applied mid-life pharmacological inhibition of the ISR, which caused similar longevity effects like the ISR mutants *ppp-1*, *gcn-2*, and *pek-1*. Importantly, the authors ruled out that *Gcn(-)* longevity is not dependent on the attenuation of global translation but is rather dependent on the upregulation of specific mRNA transcripts. A following transcriptome approach identified several mRNA transcripts that are differentially translated in hypoactive ISR worms. Further genetic experiments conducted by the authors resulted in the finding that the predicted kinase *kin-35* is fully responsible for the longevity phenotype of hypoactive ISR mutants (*ppp-1*), implying a fundamental role of *kin-35* in cellular processes that regulate longevity through the modulation of ISR.

While the link between ISR and longevity/proteostasis has been shown before in yeast and other model systems, the described findings might be of relevance to further understand the physiological role of the ISR in the aging process. The most interesting part of the data addresses how ISR inhibition affects longevity, however, it remains incomplete. The authors should address the role of the identified kinase *KIN-35* in more detail to strengthen the mechanistic novelty of their findings.

Our response:

We thank the reviewer for highlighting the strengths and novelty of our findings. We would like to point out that:

- 1. published work has shown that the activation of the ISR and, subsequently, GCN4/ATF4, can extend lifespan, particularly in yeast (Postnikoff et al., 2017 PMID: 29167799; Hu et al., 2018 PMID: 30117416), or in dietary restriction in worms and flies (Rousakis et al., 2013 PMID: 23692540; Kang et al., 2017 PMID: 27979906). Thus, while the link exists, our observations provide completely new data showing that the **inhibition** of the ISR is beneficial. Also, our data do not dispute the role of GCN2 and ISR activation in DR longevity. Our findings move the needle in aging research as they provide evidence for a yet unknown mechanism that extends lifespan without limiting protein biosynthesis. Of note, this is in line with various observations that demonstrate a protective effect of ISR inhibition in mammals (Zhu et al., 2019 PMID: 31727829, Chou et al., PMID: 28696288), potentially making the worm a useful model for the human ISR.*
- 2. Our paper is based on a genome-wide chemical mutagenesis screen that goes beyond any previous longevity screen in *C. elegans*. It enabled the identification of longevity alleles and generated both loss- and gain-of-function mutations in contrast to past RNAi screens (Hamilton et al., 2005 PMID: 15998808; Hansen et al., 2005 PMID: 16103914).*
- 3. Results from the screen show a surprising and fully unbiased convergence on the regulation of eIF2 as a key event in *C. elegans* lifespan extension and protein homeostasis.*
- 4. While the *ppp-1* mutants reported in our manuscript extend worm lifespan by modulating mRNA translation initiation, this occurs without reducing protein synthesis. Instead, we find the requirement for some genes that are regulated by selective translation.*
- 5. Finally we show that multiple targeted genetic modifications and a pharmacological treatment inhibiting the ISR extend lifespan, expanding our findings beyond the *ppp-1* mutants identified in the screen.*

Having realized that these strengths of the paper were not appropriately highlighted in the first submission, we have rearranged the manuscript now to add substantial new data and to more strongly emphasize the novelty of our findings.

Major points and concerns:

Interestingly the authors suggest that the lifespan extension caused by ISR inhibition is not mediated by overall changes in protein translation rate. Instead, they identified enhanced translation of certain mRNAs including kin-35. Besides kin-35(RNAi) mediated suppression of the extended lifespan phenotype of ppp-1 mutants it would be interesting to test the role of the kinase activity of KIN-35. Therefore, the authors should use kinase inactive kin-35 mutants to address genetic suppression of double mutants. If the kinase activity is required it would be interesting to test direct regulation of the ISR biochemically. Moreover, it would be interesting to test if KIN-35 overexpression shows lifespan extension.

Our response:

To address this important point, we have generated 3 independent kin-35 overexpressor lines and find that, while kin-35 was required for the longevity by ISR inhibition, overexpression was not sufficient to extend lifespan (Fig. 3g and h). This was not per se surprising as many key factors that are required in other longevity pathways, are not sufficient to extend survival. DAF-16 overexpression, for example, leads only to a mild lifespan increase that does not phenocopy daf-2 longevity that is fully daf-16 dependent (Henderson and Johnson 2001 PMID: 11747825). Not much is known about kin-35 and, while interesting, we consider it out of the scope of the paper to perform the full biochemical in vitro analysis of KIN-35 that would be necessary to (i) ask if it is a functional kinase and (ii) identify the catalytically active residues. While our paper finds many phosphorylation-related genes among the translationally altered mRNAs (Fig. 3c), we do not conclude that kinase activities are ultimately responsible for the lifespan extension we observe. Instead, we use kin-35 dependence, at multiple points in the paper, as strong evidence that it is the inhibition of the ISR with its consequent effects on selective translation that extends survival. All of these aspects are novel and shed a new light on the role of the ISR in aging. We have rearranged the paper to appropriately position our data on kin-35 as a functionally relevant, while not fully understood, player in ISR inhibition.

F2F: The dynamic range of the tunicamycin developmental assay is low and scarcely allows the claimed statement that eIF2aS51A mutants are hypersensitivity compared to WT worms without a statistical evaluation or stated p-value. Increasing the dynamic range of the assay might be achieved with lower tunicamycin concentrations (0.1 µg/ml, 0.5 µg/ml, 1 µg/ml). The negative control should be conducted with the solvent in which tunicamycin is diluted (e.g. DMSO). In case DMSO was not used as the solvent, a clarification in the material and methods about the solvent should be added.

Our response:

We fully agree with this point and have provided a new dose response assay with the appropriate analysis in Fig. 6b. The eIF2aS51A mutants are extremely sensitive to low TM concentrations (1, 1.5, and 2 µg/mL). It is now clarified that DMSO was used as a solvent and it is the control in the experiment.

F2G: The hypoactive ISR strain eIF2aS51A is long-lived under non-stress conditions. Following the narrative of the manuscript (discussion), the lifespan should be drastically reduced in eIF2aS51A worms under conditions where ISR activation is crucial for reestablishing cellular homeostasis (amino acid starvation, ER-stress e.g. tunicamycin treatment, viral infection e.g. orsay virus infection). This information would contribute to a clearer and better understanding whether the eIF2aS51A strain behaves according to the narrative of the manuscript and whether chronic ISR ablation is detrimental under stress conditions.

Our response:

We thank the reviewer for this suggestion that addresses one of our key thoughts behind this paper: The ISR is certainly protective under some conditions, and we believe that this is the case under external stress. At the same time, we found that in the absence of stress, promoting selective translation by ISR inhibition was beneficial. To address this point experimentally, we have performed a survival analysis in the presence of TM at a concentration that is lethal to WT development (20 µg/mL). Adult worms are generally more resistant to TM compared to developing larvae, and we find that WT animals have a mean lifespan of about 12 days in this condition. Consistent with our previous results, eIF2aS51A mutants were long-lived without TM, showing a mean lifespan extension of 24% compared to WT controls (Supplementary Fig. 6d). Adult TM treatment reduced the survival of WT animals by 30% and, consistent with our expectations, eIF2aS51A lifespan was even further reduced by TM (-44% in mean lifespan). We conclude that (i) eIF2aS51A mutant lose their longevity when confronted with TM, (ii) ISR defenses of the WT might be insufficient to protect from TM toxicity as WT and eIF2aS51A animals have the same lifespan when treated with TM.

Minor points and concerns:

Non-breaking hyphens should be used for allele names.

Our response:

The manuscript has been edited accordingly.

For better transparency about the quality of data, it is necessary to show each data point in the bar plots instead of the average + SEM. The visualization of each datapoint across independent experiments allows the reader to validate the robustness of the underlying biology. The data visualization like in Fig1I should be adapted throughout the whole manuscript for the bar plots.

Our response:

The manuscript has been edited accordingly.

F2C: Quantification of GFP/TUB values seem not proportional to the shown western blot and the fluorescent microscopy pictures in Extended Data F2B.

Our response:

The fluorescent images (now in Supplementary Fig. 4c) are each representative of multiple independent experiments and apparently the fluorescent visualization does not lead to the same large apparent difference as the Western blot. While the data are fully consistent, we have re-arranged the figures to reflect this.

F2D: The blot could have a slightly higher exposure for better visibility. The representative blot is not reflecting clearly the quantification. The Y-axis labeling [%] of the quantification is missing. Total eIF2A (unphosphorylated) levels are not shown throughout the whole manuscript. In case that there is no available antibody for total eIF2A in the worm, the genetic data in the eIF2aS51A strain is slightly compensating for this drawback, making it a minor instead of a major concern.

Our response:

We have repeated the Western blots and have included multiple additional controls to better understand the consequences of the kinase mutations from the screen. The data are now in Supplemental Fig. 4h and we have of course made sure that the data are properly labeled.

As raised by all reviewers, the total eIF2A levels are not shown throughout the whole manuscript. This is due to the lack of a reliable antibody in worms. To address this problem, we decided to detect the total eIF2 levels by quantitative proteomics based on TMT labelling. Please find representative quantifications of the TMT intensities measured for the total eIF2A and other subunits below. From this, we conclude that normalizing the eIF2 α signal to tubulin is appropriate in the manuscript.

F2I: The values of the pelf2a/TUB quantification of seem not proportional to the shown western blot.

Our response:

We thank the reviewer for pointing this out. We have a total of 7 replicates of this experiment and these are averaged in the bar graph (now Fig. 5c). While the use of two drug treatments in independent experiments leads to some variability in the results (now better depicted in the bar graph), the Western blot we show is indeed representative of the experiments we have performed.

F3B: The negative control *rks-1(sv31)* shows a robust decrease in overall puromycin incorporation and therefore global cellular translation. It would be highly desirable to repeat the experiment ones more to increase the number of independent experiments from $n=2$ to $n=3$. Especially since this positive control beautifully shows the efficacy of this assay.

Our response:

*We thank the reviewer for pointing this out, particularly as the reductions in protein synthesis of the long-lived TOR pathway mutants are indeed staggering. We have repeated these experiments and have also added new data from a mutant with reduced protein synthesis (*iftb-1*) that was identified in our screen. Further, we added a long exposure to highlight the differences in signal intensity (now Fig. 2)*

F3C & F3E: An overlay of the polysome profiling data should increase comparability between the strains (colors, transparent lines). In case that all three traces share exactly the same flow, the chosen representation might be better.

Our response:

*We provide an overlay of the polysome profiles here, demonstrating that WT and *ppp-1* mutants have identical profiles. However, we want to avoid presenting this display of the data in manuscript, as each profile is unique due to the sucrose separation.*

Extended Data F3: Are total eIF2a levels regulated in D6 worms? Is there a difference between WT and *ppp-1* strains in total eIF2a levels?

Our response:

We have now used a quantitative proteomics approach based on TMT labelling to quantify total levels of eIF2 subunits by. The data is included in the answer above on page 3.

Fig4: The elimination process of the other target mRNAs after Fig4E is non-transparent. In case that the authors tested the remaining 5 targets for their longevity effect, an additional statement and the extended data of the lifespans would be appreciated.

Our response:

*We agree that the elimination process was not sufficiently explained in the first version of the manuscript. Now we clarified the elimination process in the main text. In addition, we added a cartoon depiction of the elimination process (Supplementary Fig. 3a). As the 5 mentioned targets did not suppress the motility of both *ppp-1* mutants, they were not considered for longevity assays. Thus, the elimination process was rational and the manuscript now reflects this.*

Fig4F & G: legend is overlapping between different genotypes, which is misleading

Our response:

We have modified the figure legend accordingly.

Reviewer #2 (Remarks to the Author):

In this manuscript the authors studied the link between Eif2B/Eif2 kinases and longevity in *C. elegans*. The authors show data where eIF2B extend life span through ISR inhibition and independently of global protein synthesis. Alternatively, the authors suggest that it is dependent of the Kin35 kinase which is preferentially translated under these conditions.

In this manuscript the authors provide a solid dataset of genes that may be interesting to investigate during aging in future studies. Despite that the involvement of the eIF2 pathway in aging was nicely shown in other species (Ref 16-novelty??). Moreover, the authors fall short to show some mechanistic directions. The authors do suggest the involvement of Kin35 but not really show mechanistically how this is done and what is the role of Kin35 in longevity. Moreover, they didn't show how Kin35 is translationally enhanced. These are major concerns that prevent me to recommend this manuscript for publication in Nature comm.

Our response:

We appreciate the reviewer's summary of our paper highlighting the role of ISR inhibition in lifespan extension without reduced protein biosynthesis. Also, we would like to address and clarify the reviewer's main criticism.

*1. The eIF2 pathway has been linked to lifespan extension, but all published literature links an **activation** of the ISR to extended lifespan, for example in yeast (Postnikoff et al., 2017 PMID: 29167799; Hu et al., 2018 PMID: 30117416) and in dietary restriction-mediated longevity in worms and flies (Rousakis et al., 2013 PMID: 23692540; Kang et al., 2017 PMID: 27979906). This is consistent with stress response activation leading to enhanced robustness in a hormesis-like scenario. The novelty of our paper is that it is the **inhibition** of the ISR and the subsequent alterations in selective translation that extend lifespan. This is novel and has never been reported before. Our data do not contradict published work linking reduced translation and ISR activation to lifespan extension because we demonstrate that the change in selective translation is responsible for lifespan extension. Importantly, this is in line with various observations that demonstrate a protective effect of ISR inhibition in mammals (Zhu et al., 2019 PMID: 31727829, Chou et al., PMID: 28696288).*

2. Reference 16 (Halliday et al., 2017 PMID: 28430857) identifies compounds that counter eIF2 α phosphorylation and uses mouse models of neurodegeneration. While the findings are consistent with our observations, our manuscript provides significant novelty regarding (i) extension of normal lifespan in the absence of genetic disease models and (ii) by showing that changes in selective mRNA translation are involved in the extended lifespan. Thus, our paper provides unique and novel insights.

3. We have further investigated kin-35 and found that, while necessary for longevity under ISR inhibition it is not sufficient. Clearly, more work needs to be done to understand how kin-35 affects survival, but this goes beyond the scope of this manuscript that describes (i) a major genetic screen that was never done before at this scale, (ii) the unbiased identification of multiple eIF2 regulators and (iii) a novel mechanism of lifespan extension that alters translation of specific target mRNAs without repressing allover translation. We have rearranged the figures of the manuscript to appropriately highlight these strengths. Kin-35 is used at multiple instances in the manuscript to demonstrate its position downstream of the eIF2 pathway.

Other comments

1. Fig1, they investigated 101 genes of 127, what is the fate of the rest?

Our response:

We have rearranged Figure 1 to make this clearer. In essence, of the 127 strains, only 101 could be successfully cultivated, and thawed after freezing. This loss of mutants in mutagenesis screens is normal. 101 long-lived strains resulted from the screen and all of them were sequenced.

2. Figure 1h is shown at day 1 when eIF2 is apparently not induced in WT, what happens if they repeat the experiment in day 6 where eIF2 α is active and is induced compared to day 1 as in extended figure 3a.

Our response:

We have repeated the heat resistance assay with ppp-1 mutants and WT animals on day 6 of adulthood and observed a generally higher tolerance to heat in comparison to day 1, and we found no difference between WT animals and the mutants at day 6. Possibly, at day 6, the strong eIF2 α phosphorylation provides a protection from heat stress to WT worms and the ppp-1 mutants alike. In general, the link between aging and heat resistance is not well understood as on one hand the expression level of HSP genes are elevated during ageing (Walther et al., 2015 PMID: 25957690), on the other hand, the downstream signaling seems impaired (Kourtis et al., 2011, PMID: 21587205).

3. Everywhere where the author show eIF2 α phosphorylation this should be normalized to total eIF2 α /tubulin instead of Tubulin alone as shown in several places throughout the manuscript.

Our response:

We fully agree with the reviewer. However, in C. elegans the total eIF2 α antibodies are not reliable and so in the field it is accepted to normalize to a house keeping gene. To test if normalizing to tubulin is permissive, we have done a proteomics-based quantification of the eIF2 subunits that we show below. We find no differences between the treatments, suggesting that in this case normalizing the Western blots to the tubulin signal is justified.

4. Figure 2a the dashed lines and the colors are hard to follow.

Our response:

We have modified the Figure to make the legend clear.

5. In Fig1 and Fig2 the WBs are done with DTT while the survival assay is done with Tm, lack of consistency. What is the status of eIF2 α -p under Tm ?

Our response:

We have repeated many of the stress experiments and consistently find a similar response in DTT or TM treatments. We have re-arranged the figures to show consistent treatments.

6. The concentrations of Tm are very high and kill most of the WT more than 70% with the lowest concentration that was used. Tm toxicity can be unspecific, I wonder what happens if they repeat this with lower concentrations of Tm that are enough to phosphorylate eIF2 α and then to compare % of adult.

Our response:

We agree that the TM dose response assays lacked some lower concentrations. We have now repeated this experiment and provide the new data in Fig. 6b. The TM hypersensitivity of the eIF2 α S51A point mutant was confirmed.

7. What is the status of the other UPR arms in these mutants? Is there a compensation of the other pathways that cause miRNA degradation that in turn stabilize Kin35 and enhance translation?

Our response:

We thank the reviewer for this suggestion. First, we have used RNAseq and qRT-PCR to quantify downstream targets of the ER-UPR and found no differences between the *ppp-1* mutants and WT animals when we compared baseline expression or during a TM challenge (Supplementary Fig. 4d and e).

Second, to address the possibility of miRNA regulation, we used two independent software tools for miRNA target site predictions (TargetScanWorm Release 6.2 and PicTar). While both detected miRNA target sites in known miRNA target genes such as *lin-14* (Shen et al., 2012 PMID: 23239738), for *kin-35* no miRNA target sites or only sites with poor conservation could be detected, making miRNA mediated regulation of *kin-35* unlikely.

PicTar WEB INTERFACE

Choose Species:	nematode
Choose Dataset:	single microRNA target predictions
microRNA ID: <small>Click above for all microRNAs linked to RFAM</small>	cel-let-7
Gene ID: <small>Click above for all wormbase Id's linked to WormBase (Warning: may take ~20 secs)</small>	M04F3.3 <small>nematodes: use wormbase identifiers (for example F13D2.1 or par-6). Currently, only exact matches are supported. searches are case insensitive.</small>

There are no predictions for this gene.

PicTar WEB INTERFACE

Choose Species:	nematode
Choose Dataset:	single microRNA target predictions
microRNA ID: <small>Click above for all microRNAs linked to RFAM</small>	cel-let-7
Gene ID: <small>Click above for all wormbase Id's linked to WormBase (Warning: may take ~20 secs)</small>	lin-14 <small>nematodes: use wormbase identifiers (for example F13D2.1 or par-6). Currently, only exact matches are supported. searches are case insensitive.</small>

```

19900_Ele_T25C12.1a AATGCCAATTTTCG-----AGTCATCCTTCG--GGCAATGTT-----CATFACACTTTCCTCTGTTG--TACTTGAGCATGTTCAATTTCAAT-----
19900_Bri_T25C12.1a ATTG-----TTTCGCCCAACATCGTCTTCCAATGAGCAGC-----GTCCACA-----TTATCTGTTGTTATAGTGGAGCATGTTGGACTTTCAGTT-----
19900_Rem_T25C12.1a AATG-----T-----CAACAT-GTCAGCCAATGAGCAGCAGTCTTATGCCCCACCACCATCCTACTCTGTTG--TACTTGAGCCTGTTCAATTTCAATTTCCGCGGT
      10          20          30          40          50          60          70          80          90         100         110
19900_Ele_T25C12.1a -----TACTTTGT-----AAGTCCGTTTACTGCGCCCAATTCCTCGTCATTTTGATTACACTCTCTTTTAACTCAACTCAGG
19900_Bri_T25C12.1a TTTTTCCTTTCCTCTATATTTTGTTCCTATTCACAATGCTCTCTTTTCTTACCTTCCGACTAGTCCCAAAATTCCTCGTCATTTTCGATTACTC-----CAAACCAACTCAGG
19900_Rem_T25C12.1a TTATTG-----TTTTATATCCAAATGTTCT-----TTTAAATTCGATTAGTGTGCCCAATTCCTCGTCATTTTCGATTACTC-----TA-TCCCAACTCAGG
      1010         1020         1030         1040         1050         1060         1070         1080         1090         1100         1110
19900_Ele_T25C12.1a TCCATCTTAACATCCATCCCATTTGACCTCTGAATCTTGCTTCGCTTACCTCGTAACAATATATTTTTATCGGCTTAAACCTAATAAATCATTTACCAGAAAAACATTGT---AGC
19900_Bri_T25C12.1a TCCAAATT-ACAGTCCATCCCAATGACCTCTGAATCTTGCTTCGCTTACCTCGGAATTGAA-----TATCGGCTTAAATTCATAAAAACCATTTAC-ATCTTAATTTTGTTTGATA
19900_Rem_T25C12.1a TCCGACT-ACCATTCATCCAATTTACCTCTGAATCTTGCTTCGCTGACCTCTATTTT-----CATCGGCTTAAATTTAATAAAAACATTTA-----
      2010         2020         2030         2040         2050         2060         2070         2080         2090         2100         2110

```

PicTar score for cel-let-7:15.4658 Ele:21.7321 Bri:10.2703 Rem:14.4414

Org	PicTar score	PicTar score per species	microRNA	Probabilities	Nuclei mapped to alignments	Nuclei
Ele	15.4658	21.73	cel-let-7	0.99 0.93 0.90 0.93 0.93 0.90 0.93 0.86	279 560 858 1168 1182 1332 1643 1979	222 423 652
Bri	15.4658	10.27	cel-let-7	0.84 0.84 0.97 0.84 0.73	560 1168 1407 1780 1979	378 9
Rem	15.4658	14.44	cel-let-7	0.90 0.83 0.87 0.90 0.83 0.90 0.83	560 847 858 1168 1183 1780 1970	461 656

C. elegans M04F3.3 172091.0 3' UTR

C. elegans lin-14 181337.1 3' UTR [Other transcripts]

8. In *C. elegans* most of the UPR signaling was shown to occur through IRE1 whereas PERK and ATF6 pathways were modestly involved (initially reported in PLoS Genet. 2005 Sep;1(3):e37), how do the authors reconcile their data with these results especially in the context of UPR induction?

Our response:

While IRE-1 plays a key role in UPR signaling in the nematode, the other kinases are functional and involved in important signaling events. We use ER stress merely as a means to trigger eIF2 α phosphorylation in our experiments that probe ISR induction. The IRE-1 pathway of ppp-1 mutants appeared to have a normal response to ER stress (Supplementary Fig. 4d and e). Thus, these other pathways appear unaffected by the inhibition of the ISR. We find that ER stress is appropriate and sufficient to induce the eIF2 α phosphorylation and the ISR and thus use it as a tool to delineate the ISR in our novel mutants.

9. Instead of blocking eif2a phosphorylation it would also be of interest to evaluate how maintaining its phosphorylation through the use of compounds such as Sephin1, Salubrinal or guanabenz (the last 2 having already been used in *C. elegans*) would affect the observed phenotypes (aging/longevity).

Our response:

*We fully agree with this comment and have addressed this extensively. However, the worm emerges as an inappropriate model for these experiments, preventing clear conclusions. We have supplemented salubrinal and guanabenz at various concentrations and did not detect an activation of the ISR using our *atf-4P::GFP* ISR reporter strain (see below). At higher doses (100 and 200 μM), nematodes looked sick after the guanabenz treatment. We nonetheless performed lifespan assays at 25 μM , a concentration that has been used in *C. elegans* (Fardghassemi et al., 2017; PMID: 29061563), and found no effect on survival. We further used ISRIB and could likewise not detect an effect on the ISR (Supplementary Fig. 5b). We think that either *C. elegans* is very effective in eliminating these substances, which is a known general feature of the worm, or that the target proteins show lower affinity to the compounds that were developed in higher organisms. To more carefully address this point genetically, we have generated an *eIF2 α* phospho-mimic S51D (Fig. 6 and Supplementary Fig. 6). This substitution is developmentally lethal in homozygous animals and heterozygous mutants show a short lifespan, low fecundity, and retarded development, suggesting that a chronic ISR is detrimental in the worm.*

10. Title is misleading, how eIF2B does extend lifespan? Is it its activity that is important? not clear from the title.

Our response:

We thank the reviewer for pointing this out. We have decided to change the title to better focus on the conceptual novelties of the paper.

Reviewer #3 (Remarks to the Author):

In this manuscript, Derisbourg and colleagues investigate the role of integrated stress response (ISR) in regulation of ageing. A core event of ISR is the phosphorylation of eukaryotic initiation factor 2 α (eIF2 α) by one of four eIF2 α kinases, i.e. PERK, PKR, HRI and general GCN2. In an effort to identify novel longevity modulators, the authors performed an unbiased forward screen by using 0.3% ethyl methanesulfonate (EMS) in a conditionally sterile *C. elegans* strain. The screen revealed that mutations in the *ppp-1/eIF2By*, *pek-1/PERK* and *gcn-2/GCN-2* genes were able to extend lifespan and confer protection against proteotoxic stress caused by heat shock and the accumulation of aggregation prone proteins such as polyQ35. These data suggest a causative relationship between ISR and longevity. Moreover, an engineered phospho-defective eIF2A α S51A mutant was found to be long-lived and heat resistant. Consistently, pharmacological inhibition of ISR by estradiol valerate reduced GFP induction that occurs in animals expressing the *atf-5P::GFP* reporter after tunicamycin treatment. By contrast, ISR induction with propafenone hydrochloride further increased GFP expression in this reporter strain and shortened the lifespan of wild-type animals. *ppp-1* mutants exhibited comparative protein synthesis rates as the wild-type controls as evidenced by measurements of incorporated radioactive methionine and surface sensing of translation (SUnSET). In addition, polysome profiling failed to detect any difference in the overall distribution and abundance of wild-type and eIF2A α S51A mutant animals. Together, these data suggest that the extended lifespan of *ppp-1* mutants is uncoupled from reduced global protein synthesis. Further analysis of polysome-associated mRNAs showed that genes encoding proteins involved in phosphorylation events are enriched in *ppp-1* polysome fractions compared to wild-type controls. Interestingly, translational efficiency of specific mRNAs, including *kin-35* was increased in *ppp-1* mutants. Knockdown of *kin-35* suppresses the enhanced longevity of *ppp-1* and eIF2A α S51A mutant animals and abolished the motility of polyQ35; *ppp-1(wrm10)* animals. These findings suggest that *kin-35* is required for lifespan extension and increased proteostasis in animals carrying the above mentioned *Gcn(-)* class mutations. This manuscript presents interesting findings which suggest a role for ISR in regulation of ageing. However, there are several experimental and interpretation issues that need to be further addressed.

Major concerns

The title of the manuscript pertains to the mechanism by which eIF2B enhances longevity. This mechanism seems to rely on the inhibition of ISR. The authors have attempted to inhibit ISR in a pharmacological manner by using estradiol valerate. This compound (as every compound used in a living organism) can have pleiotropic effects, especially if it is a steroid hormone with the potential to activate nuclear hormone receptors (there are numerous in *C. elegans*). There are other available chemical compounds in the literature that can inhibit ISR specifically. The most prominent example is ISRIB which enhances the assembly of the eIF2B complex (doi:10.1111/febs.15073). ISRIB is a drug-like small molecule that suppresses ISR (Costa-Mattioli and Walter, 2020, Science, 368, 384) and binds to eIF2B leading to its activation (Zyryanova et al, 2018, Science, 369, 1533-1536).

Our response:

*We agree with the reviewer's assessment and have used, in addition to the strong genetic evidence in the manuscript, pharmacological modulators of the ISR. We had of course first tried ISRIB, which works very well in our hands using mammalian cells in other projects. In worms, however, we see no effect of ISRIB using the *atf-4::GFP* ISR reporter (Fig. 5a), or on survival, an observation we now include in the paper to clarify this point (Supplementary Fig. 5b). *C. elegans* are known to effectively eliminate foreign compounds, potentially preventing the accumulation of sufficient cellular ISRIB concentrations. Alternatively, given the amazing fit of ISRIB in a cleft of the mammalian eIF2B complex that is not fully conserved in the worm, ISRIB might not have the same high affinity to *C. elegans* eIF2B.*

*While we cannot rule out pleiotropic effects of estradiol valerate, we show evidence for its activity on the ISR, and we show that the downstream component *kin-35* is required for the estradiol valerate effect on longevity (Fig. 5d). Furthermore, we demonstrate that estradiol valerate does not further increase lifespan of *ppp-1* animals suggesting that estradiol valerate might target ISR components (Fig. 5e). Moreover, the fact that other estradiol esters (estradiol acetate, estradiol diacetate, estradiol methylether, estradiol propionate, estradiol 3-sulfate) did not have an effect on the ISR lets us conclude that estradiol valerate is specific in its effect on the ISR (Halliday et al., 2017 PMID: 28430857).*

Finally, we have decided to alter the title of the paper to better highlight the conceptual novelties of the paper.

Two very interesting studies published recently provided intriguing findings about an OMA-1-DELE-1-HRI axis which relays mitochondrial stress to the ISR (Guo et al, Nature 2020 and Fessler et al, Nature 2020). The authors have exclusively examined ISR activation upon administration of DTT, an ER stressor. It would be nice if they could extend some of their observation to mitochondrial stress (such as CCCP, oligomycin or antimycin).

Our response:

*Thanks for this comment, we were also intrigued by the recent papers and wondered if this pathway might exist in the worm, which only has two eIF2 kinases, GCN-2 and PERK/PEK-1, thus lacking the molecular link via HRI. We are fairly certain that *gcn-2* and *pek-1* are the only eIF2 α kinases as the double KO has no residual eIF2 α phosphorylation in Western blots. Nonetheless we analyzed ISR activity upon mitochondrial stress. The mitochondrial stress reporter *hsp-6::GFP* was induced by antimycin (Ant A) and paraquat (Pq) treatments (see below). eIF2 α phosphorylation remained unchanged with antimycin and paraquat treatments, but was increased by DTT (see below). Thus, even though one study demonstrated a possible link between mitochondrial stress and the*

ISR in the worm (Baker et al., 2012 PMID: 22719267), we did not observe an ISR induction upon mitochondrial stress. In fact, in the context of the paper, we merely used ER stress as a tool for ISR induction and while the source of the stress did not matter, our analyses focused on the ISR and translation initiation pathway downstream of eIF2.

The existing literature suggests that mutations in neither PEK-1/PERK nor GCN-2 extend *C. elegans* lifespan (Henis-Korenblit et al, PNAS 2010 and Baker et al, Plos Gen 2012). The second study specifically suggests that GCN-2 is only required for the extended longevity of mitochondrial mutants (like *isp-1* or *clk-1*). Since the strains used in this study originated from random mutagenesis (with the possibility of accumulation of background mutations) this discrepancy should be convincingly explained. Does genetic inhibition of these three components via RNAi also extend lifespan of wild type animals? What is the nature of the described mutations? Are they null or hypomorphs?

Our response:

The reviewer raised an important discrepancy. We agree that, as it is reported in the literature, pek-1 and gcn-2 full knockout mutants are not long-lived. Similarly, RNAi experiments abolish the full mRNA. In contrast, the gcn-2(wrm4) and pek-1(wrm7) mutations from our longevity screen carry one single nucleotide polymorphism each, which strongly reduced but did not fully abrogate the kinase function (Supplementary Fig. 4h). While we do not fully understand the difference between the KO and the point mutant alleles the underlying mutations might explain this discrepancy: the wrm4 and wrm7 alleles result in single amino acid substitutions in the ISR kinases. Thus, first, it is likely that the kinases are part of complexes that would be affected by full knockout but not by the individual point mutants. Second, while converging on eIF2, the kinases might have other substrates that affect worm stress responses. We now raise these possibilities in the discussion of the manuscript.

Do phosphomimetic mutants (ex. eIF2aS51D or eIF2aS51E) show declined protein homeostasis and lifespan? Is any phosphatase [ex. Type 1 protein phosphatase (PP1) (or other)] able to compete/dephosphorylate eIF2? In this case does PP1 regulate protein homeostasis and lifespan?

Our response:

To test this possibility, we have generated an eIF2aS51D mutant and found that the homozygous state is lethal. Heterozygous mutants are short-lived and are characterized by retarded development and low fecundity (Fig. 6h and Supplementary Fig. 6). Thus, chronic ISR activation is detrimental in worms, which is consistent with chronic ISR activation in a number of human neurodegenerative conditions.

There is very limited knowledge about eIF2a phosphatases in the worm. One study suggests the gene gsp-1 as its knockdown enhances eIF2a phosphorylation, however the activity appears to be rather broad as the paper also finds that gsp-1 RNAi severely shortens wild type survival and the authors conclude that gsp-1 depletion has "pleiotropic, non-specific effects" (Baker et al., 2012 PMID: 22719267)

Taken together, our combined experimental work illuminates the role of eIF2a through multiple independent approaches. Our evidence, using genetic and pharmacological manipulations, shows that ISR inhibition extends survival, while chronic ISR activation shortens lifespan.

Regulation of eIF2 GTPase by typical circle interventions: Can the above results in protein homeostasis and lifespan be confirmed by constitutively active and dominant negative eIF2a. What's the role of eIF2 GTPase-activating protein eIF5?

Our response:

The reviewer, to our understanding, suggests a direct manipulation of the eIF2 enzymatic activity to further solidify our results. Generating a GTP-locked version of eIF2, however, would not serve to constitutively activate the enzyme as the output of the pathway is translation initiation that requires the cycling of eIF2 through GTP and GDP-bound states. Locking eIF2 in the GTP-bound state would thus lead to massive defects in translation initiation. The optimal means to activate eIF2, as used in the paper, is the generation of the S51A substitution in the α subunit that prevents the inhibitory interaction with eIF2B.

eIF2 GTPase activation by eIF5 overexpression would be expected to enhance eIF2 activity and translation initiation, potentially mimicking the dominant eIF2B mutations. However, the interaction of eIF5 with eIF2 and eIF2B appears to be more complex and overexpression of eIF5 results in reduced eIF2B activity and reduced ternary complex activity (Singh et al., 2006 PMID: 16990799) and eIF5 overexpression leads to ATF-4 induction in human cells, suggesting an inhibition of eIF2 and induction of the ISR (Kozel et al., 2016 PMID: 27325740). Together, these data suggest that eIF5 manipulation would have severe side effects and would not appropriately mimic eIF2B activation or enhanced eIF2 turnover. Given this complexity, we decided against the overexpression of eIF5.

Given that activation of ISR has been associated with several pathological conditions, such as neurodegenerative diseases, diabetes and metabolic disorders, among others, it would be of interest to investigate how manipulation of ISR in specific cell types, for example in specific nematode neurons, can influence whole organism survival.

Our response:

We are in full agreement with this suggestion and are planning such experimentation for the future. This possibility is further supported by our work using toxic proteins that are expressed in specifically in muscle tissue. However, the tissue specific function of ISR inhibition as well as the potential inter-organ communication controlling the ISR go beyond the scope of this paper. In this study we focus on the very large genomic screen for longevity, the convergence on ISR regulators, the finding that our mutants have extended lifespan without reductions in allover

translation, and the strong additional genetic and pharmacological evidence that supports the notion that ISR inhibition extends lifespan.

The fact that heterozygous *ppp-1* mutants are long-lived suggest that the *ppp-1(wrm10)* and *ppp-1(wrm15)* mutations are genetically dominant. However, it is not clear to me how the finding that *ppp-1* RNAi abolishes longevity of *ppp-1* mutant animals supports the abovementioned conclusion (please, see lines 90-92, page 4).

Our response:

*We fully agree with the reviewer's assessment that the *ppp-1* mutants are dominant, based on the phenotype in the heterozygous mutants. It is, then, fully consistent that partial depletion of the *ppp-1* mRNA by RNAi would counter the dominant effect of the *ppp-1* mutations. We have rephrased the respective section of the paper to make this clearer.*

The enhanced resistance of *ppp-1* mutants to heat stress might be due to the activation of heme-regulated eIF2 α kinase (HRI) that reportedly involves the HSP90 and HSP70 proteins (Lu et al, 2001, Mol Cell Biol 21: 7971 – 7980). Did the authors check this possibility?

Our response:

*We thank the reviewer for the suggestion. While a heat shock response downstream of the activated ISR is likely, our data point in a different direction. First, conceptually, our *ppp-1* mutants inhibit the ISR, while the HRI mediated effect would be mediated by ISR induction. Second, there is no HRI homolog in *C. elegans*. Third, we did not observe significant HSP induction in our transcriptome or translome data from the *ppp-1* mutants.*

The authors claim that pharmacological inhibition of ISR by estradiol valerate reduces the expression of an *atp-5P::GFP* reporter and extends *C. elegans* lifespan. By contrast, ISR induction with propafenone hydrochloride further elevated *atp-5P::GFP* expression and shortens lifespan. However, both compounds i.e. estradiol valerate and propafenone hydrochloride have been reported to act as UPRER suppressors (Halliday et al, 2017, Brain: a journal of neurology, 140, 1768-1783). Hence, their differential effects on *atp-5P::GFP* needs further investigation.

Our response:

*We thank the reviewer for this comment and have updated the manuscript to clarify this apparent contradiction. The elegant 2017 paper by the Mallucci lab set out to screen a compound library for modulation of developmental TM toxicity and compounds that enhanced worm development in the presence of TM were scored as UPR inhibitors. Both estradiol valerate and propafenone belonged to the top hits in this screen. However, they did not counter the TM-induced expression of a *chop::luciferase* reporter in cells. We therefore decided to re-evaluate the top hits from the compound screen using the *atf-5::GFP* ISR reporter in worms. We found that estradiol valerate, consistent with Halliday et al., is an ISR inhibitor. Propafenone hydrochloride, on the other hand, further elevated TM-induced ISR signaling (Fig. 5a) and shortened *C. elegans* lifespan.*

The authors claim that estradiol valerate suppressed eIF2 α phosphorylation upon DTT treatment. However, the decrease observed is not statistically significant compared to vehicle (DMSO)-treated animals exposed to DDT as Fig. 2i showed.

Our response:

*DTT strongly elevated eIF2 α phosphorylation, but this did not occur in the presence of estradiol valerate. While the suppression of the eIF2 α phosphorylation by estradiol valerate did not reach significance due to the variabilities in working with compounds in *C. elegans*, we showed that 1) DTT is significant vs DMSO treatment and the combination of DTT and estradiol valerate is not; 2) a clear effect of estradiol valerate on worm physiology, even late in life; 3) genetic interaction with *kin-35* and *ppp-1* position the effect of estradiol valerate in the ISR pathway.*

Although ATF4 is the master transcription factor in ISR, it acts in combination with other proteins and transcription factors in certain cases and also it can be regulated at the transcriptional level by TFEB and TFE3 transcription factors during ISR. The authors need to provide further mechanistic insight into how modulation of ISR influences lifespan.

Our response:

*We thank the reviewer for this comment. We agree with the involvement of ATF4 being the most described ISR effector in mammals. However, we want to clarify that the main claim of the paper is that lifespan extension can occur in a state of ISR inhibition in which ATF-4 is not activated. Instead, the ISR inhibiting mutations in our paper actually inhibit the expression of uORF regulated ATF-4. Further, in *C. elegans* there is very little knowledge about *atf-4* transcriptional regulation.*

Among the seven genes that suppress the motility of polyQ35; *ppp-1(wrm10)* animals when knocked down, the authors chose to investigate further the effects of only two genes, C01A2.5 and M04F3.3. It appears, therefore, that the selection of genes to further examine and also the interpretation of the results is biased. For example, the D1014.3 gene encodes a protein that is reportedly involved in IRE-1-mediated unfolded protein response. This gene as well as the other ones that have been shown to be non-toxic to wild-type animals should also be further tested for their effects on longevity and the maintenance of proteostasis in *ppp-1* mutant animals.

Our response:

We thank the reviewer for this comment. The seven genes that suppress motility in polyQ35; ppp-1(wrm10) animals were found in the RNAi suppressor screen (Fig. 3d). In order to validate our results, we performed more quantitative motility assays (body bending in liquid) in both ppp-1 mutants and found that from the seven genes, only two significantly affected motility of both mutants: C01A2.5 and M04F3.3. D1014.3 was only reducing the motility of the ppp-1(wrm10) allele, thus, it was not included in further experiments. We understand that our description of the selection process was incomplete, so we changed the according section in the manuscript and added a graphical cartoon in Supplementary Fig. 3a) to improve the explanation.

More importantly, there are proteins that show very high amino acid similarity to the so-called KIN-35 protein based on blast analysis. The authors should examine whether selective translation of the corresponding mRNAs (for example, W09C3.1, T11F8.4) is also required for lifespan extension and increased homeostasis in *Gcn(-)* mutants.

Our response:

We thank the reviewer for this suggestion. We have analyzed our polysome sequencing data and determined that W09C3.1 and T11F8.4 are not subjected to selective translation (see graph below).

Authors show that KIN-35 is essential for inhibited ISR-mediated lifespan extension. To see if KIN-35 can sufficiently enhance longevity they must show that KIN-35 overexpression in wild type background can increase lifespan. Also, tissue-specific overexpression of KIN-35 can shed light on the mechanism through which this protein affects lifespan. For example, does KIN-35 affect lifespan through systemic or cell autonomous effects? Also, more experiments are required to prove that KIN-35 improves proteostasis, such as analysis of reporter systems known to be affected by proteostatic factors.

Our response:

*We thank the reviewer for this comment. We generated three independent kin-35 overexpressor lines. However, we did not observe increased longevity demonstrating that kin-35 is not sufficient to promote longevity. In line with this, kin-35 overexpressor lines do not display increased thermo-resistance. As stated above, this was not per se surprising as many key factors that are required in other longevity pathways, are not by themselves sufficient to extend survival, such as the FOXO family transcription factor *daf-16* in the longevity mediated by reduced insulin signaling.*

Several reports show that different levels of ISR alterations can have different effects on homeostasis and survival (Pakos-Zebrucka, 2016, EMBO reports). Excessive increase of ISR can lead to cell death, while mild increase improves survival. The authors must check to what extent ISR levels were affected in their experiments, e.g. through biochemical analysis of proteostatic markers.

Our response:

*These data are indeed intriguing as they suggest various outputs of the activated ISR. We would like to point out, however, that ppp-1 mutants show longevity in a state of ISR inhibition, which was fully supported by the eIF2 α S51A mutant in which the key phosphorylation event of the ISR is disabled. Thus, our findings do not fall within different levels of ISR alterations, as the reviewer suggests, but instead constitute an ISR inhibition. To address the effect of the ppp-1 mutations on stress-induced ISR signaling, we exposed our mutants to a range of tunicamycin concentrations (Fig. 4c and d). Our data using the *atf-4::GFP* reporter indicate that the maximum ISR induction is reached at a tunicamycin concentration of 50 μ g/mL and *atf-4* induction could not be further boosted*

using higher concentrations (we did not exceed concentrations of 100 µg/mL tunicamycin as higher concentrations make the worms very sick). ISR inhibition mediated by the *ppp-1* mutations reduced the *atf-4::GFP* reporter signal at all TM concentrations.

In animal models it has been shown that, when activity of genes regulating protein synthesis is altered, feedback mechanisms affect insulin signaling, hence longevity. To clarify this, authors must measure activity of insulin signaling in the whole body of the animals, or in a cell autonomous way.

Our response:

*We thank the reviewer for this suggestion. We evaluated the activity of the insulin signaling pathway with two readouts. First, we performed dauer formation assays (if the insulin pathway is downregulated, animals are more likely to enter the alternative developmental dauer state during harsh conditions such as heat stress during development). Second, we measured the mRNA levels of FOXO/daf-16 target genes. From these two independent methods, we conclude that the *ppp-1* animals do not display any change of insulin signaling pathway activity. The data are now provided in Supplementary Fig. 4f and g.*

It has been reported that, when ISR is at a very high level, ISR inhibitors are insufficient to reduce ISR-induced cytoprotection (Costa-Mattioli et al., 2020). The authors must show that the results of ISR-inhibitors treatments are independent of the stress levels of the worms prior to their treatment.

Our response:

*We thank the reviewer for this important suggestion. We as well have appreciated the corresponding literature such as the Rabouw et al. 2019 paper (PMID: 30674674) with great interest. In human cells, ISRIB activity occurs in a defined window of ISR activation, based on the stoichiometry between eIF2 and eIF2B. In intact worms, we observed that the ISR remains inhibited in *ppp-1* mutants even when tunicamycin-induced ER stress was very high (Fig. 4c and d). Nonetheless, baseline ISR levels in *ppp-1* mutants do not differ significantly from WT animals (Supplementary Fig. 2c), suggesting that the ISR inhibition upon tunicamycin treatment occurs independently of the levels of prior ISR induction.*

Minor comments

More total protein measurement methods (such as western blot analysis with more than one control antibodies and fluorescent reporters) must be applied to further indicate that total proteins levels are not affected by reduced ISR.

Our response:

*We thank the reviewer for this comment. Besides demonstrating with multiple techniques (S35-Met incorporation, puromycin incorporation, and polysome profiling) that global translation is unchanged the *ppp-1* mutants, we added puromycin incorporation measurements in the eIF2αS51A mutant (Supplementary Fig. 6e). All these data clearly demonstrate that protein levels are not affected by reduced ISR.*

The authors observed increased motility of polyQ35 expressing nematodes upon ISR inhibition. Are the polyQ35 aggregates less in *ppp-1(wrm10)*, *ppp-1(wrm15)*, *gcn-2(wrm4)* and *pek-1(wrm7)* mutants? Is this a direct effect of general protein translation modulation or other cytoprotective mechanisms (e.g. autophagy or proteasome) are induced upon ISR inhibition?

Our response:

We appreciate this question as we have used the proteotoxicity models, now also adding α-synuclein toxicity (Fig. 1k), as a marker for robustness and healthspan. polyQ35::GFP provides a fairly diffuse GFP signal that makes it difficult count aggregates. A biochemical analysis of the insoluble material would be very interesting, but it would be part of a study that asks which downstream mechanisms, such as autophagy or the proteasome are responsible for the enhanced protein homeostasis, which is beyond the scope of the current manuscript.

Is ISR inhibition also protective against other models of proteostasis collapse, such as models of Alzheimer's disease (Tau-, Aβ-overexpressing nematodes) and Parkinson's disease (α-synuclein overexpressing nematodes)?

Our response:

*To address this point, we have crossed the *ppp-1* mutants into a Parkinson's disease model (*Punc-54* driven α-synuclein overexpression). As reported by the literature, WT animals display pronounced paralysis. Consistent with our polyQ35 data, the *ppp-1* animals also have an increased motility. The data are now included in Fig. 1k.*

Do pharmacological ISR inhibition and/or *ppp-1(wrm10)*, *ppp-1(wrm15)*, *gcn-2(wrm4)* and *pek-1(wrm7)* mutants display improved cognitive and muscle function during ageing?

Our response:

*To address this point, we have performed a time course to determine whether the *ppp-1* mutants display improved muscle function during ageing. We have observed that at the age of day 12, the *ppp-1(wrm15)* animals display slightly increased motility suggesting improved muscle function during ageing for this mutant only. Given technical difficulties, we decided not to validate age related cognitive declines in the worm.*

What do the dashed lines in Fig. 2a and Extended Data Fig. 2a denote? I suppose they correspond to survival curves of *ppp-1(wrm10)* and *ppp-1(wrm15)* animals subjected to *ppp-1* RNAi. The lifespan curves are not easily discriminated in Figures 2a, 4f & 4g.

Our response:

We have modified the figure to better discriminate the lifespan curves.

The authors could check the expression of additional genes and not just *atf-5* (i.e. CHOP, ATF4 and/or GADD34) known to be induced upon ISR.

Our response:

*We are in agreement with the reviewer; however, we are confronted with a less detailed knowledge of the *C. elegans* ISR in comparison with the yeast or mammalian literature. First, *atf-4* is the sole uORF regulated gene that is well described in *C. elegans*. Second, we are convinced that measuring the phosphorylation of *eIF2 α* together with the uORF regulation of the *atf-4* promoter is sufficient to precisely evaluate the state of the ISR.*

As the authors state, they use the RNAi clones from the available libraries (Ahringer and Vidal). It is crucial that at least for those presented in the figures they provide a RT-PCR quantifying the reduction of mRNA level upon RNAi treatment.

Our response:

*We thank the reviewer for this comment. We now provide the mRNA level of *kin-35/M04F3.3* after RNAi knockdown with the respective clone from the Ahringer library, demonstrating effective knockdown (Supplementary Fig. 3c). However, for the C01A2.5 RNAi clone, qPCRs would not be informative as this RNAi clone belongs to the Vidal library that used whole ORFs.*

Asterisks are missing in Extended Figure 4a.

Our response:

None of the clones used in WT animals showed a significant difference, which is now noted in the figure legend.

Correct legend in Figure 4e. WT condition is not shown in the image as described in the legend. Or maybe the Figure title is wrong, If the graphs in the figure are title correct, then why the two different mutants are differently affected by some of the genes?

Our response:

We thank the reviewer for this comment and we modified the figure legend accordingly.

Line 73: Replace “Fig. 1b” with “Fig. 1c”.

Our response:

Thanks for pointing this out. Done.

Lines 110-113 and Fig. 2f: Is the effect of 2 or 4 $\mu\text{g/mL}$ TM statistically different between WT and eIF2aS51A mutants to support hypersensitivity to ER stress?

Our response:

To provide a better dose response assay, we repeated the experiment with a narrower tunicamycin concentration range. It shows that the eIF2aS51A mutant is extremely sensitive to low tunicamycin concentrations (1, 1.5, and 2 $\mu\text{g/mL}$) and the differences are significant compared to the WT controls. We thank the reviewer for this suggestion and modified the figure and corresponding text accordingly (Fig. 6b).

Line 121 and Extended Data Fig. 2g: Control chemical treatments without TM are missing in order to assess uORF-regulated translational activation via atf-5 reporter.

Our response:

The reported data stem from our small screen of ISR inhibitors that was necessarily performed in the presence of TM. Propafenone enhanced the TM-induced ISR response and was therefore classified as an ISR inducer. Consistent with the ISR constitutive eIF2aS51D mutant, it shortened lifespan.

Fig. 3a and b: Why were different positive controls used in the radioactive methionine labeling (a) and SUNSET (b) (ifg-1 and rsk-1/S6K respectively)?

Our response:

We have now updated the respective figures and use consistent controls.

Lines 165-167 and Extended Data fig. 4a: C01A2.5 seems to have motility reduction in WT animals [on top of “shortened WT lifespan suggesting general toxicity” (lines 173-175 and Extended Data fig. 4c)]. Was that the reason at first place that C01A2.5 RNAi wasn’t included (or presented) in lifespan analyses of ppp-1 and eIF2a-S51A longevity (Fig. 4f-g)? It would make sense to mention/discuss it before omitting one of the two candidates at the immediate following experiment.

Our response:

We thank the reviewer for this comment. We clarified the elimination process of the RNAi suppressor screen in Supplementary Fig. 3a. Regarding the case of C01A2.5, we agree with the reviewer that there is a trend toward a decreased motility in the WT background. The lifespan results clearly demonstrated that the C01A2.5 RNAi treatment is toxic as it shortened the lifespan of WT nematodes. Thus, we placed the RNAi lifespan in the supplementary data.

The Authors might consider moving Fig. 3 to extended data and move some extended data to main figures (ex. Extended Data Fig. 2a, b, f, g, h).

Our response:

We thank the reviewer for this suggestion. However, we would keep this data in the main manuscript as protein synthesis is a well described and an important means of regulating aging. It is important for us to show that our phenotypes are independent of reduced translation rates. We now add further significant data showing that the initial screen also identified mutants that extend survival through reduced protein synthesis. The observation that ISR modulators can extend lifespan through fully separate means, i.e. with and without reducing protein synthesis, is a main message of the paper.

The meaning of ISR induction or inhibition throughout the ms is sometimes confusing to the reader. It is used both when referring to eIF2a/eIF2b mutations (as these two genes are critical to the ISR cascade) and to the activation/inhibition of the downstream transcription factors (GCN4 homologues) (compare lines 179 and 182). It would be clearer if the IRS inhibition/induction would refer only to the end effect (ATF-5 protein levels).

Our response:

We changed the manuscript to be clearer in separating the terminology of ISR and translation initiation. However, as it is well accepted in the literature that an activation of eIF2B with ISRIB is referred to as an ISR inhibition, we also use this term in the context of eIF2B and eIF2. Along these lines, we find it important to point out that the eIF2aS51A mutation inhibits the ISR. However, we have edited the manuscript and have also limited the use of the term gcn(-).

The authors should discuss other studies' findings, showing that increased ISR extends lifespan in yeast (Postnikoff et al., 2017, Microbe cell).

Our response:

We thank the reviewer for pointing this out and have included this with more emphasis in the discussion.

Reviewer comments, second round:

Reviewer #1 (Remarks to the Author):

While the authors addressed most minor comments to the manuscript the major point regarding the role of the identified kinase KIN-35 remained unclear. In contrast to kin-35 overexpression, which surprisingly did not cause lifespan extension (!), biochemical and genetic investigations of the kinase activity are necessary to provide mechanistic novelty for justifying publication in Nat. comm.

Reviewer #2 (Remarks to the Author):

The authors have done a good job at addressing the issues raised on the initial version of their manuscript. The paper is now much stronger and I have no further comment on the revised manuscript.

Reviewer #3 (Remarks to the Author):

The authors have adequately responded to my comments.

REVIEWER COMMENTS – 23 December 2020

We would like to thank the reviewers for their work on our manuscript, their comments have helped us strengthen the paper.

Below we will address the reviewers' final comments.

Reviewer #1 (Remarks to the Author):

While the authors addressed most minor comments to the manuscript the major point regarding the role of the identified kinase KIN-35 remained unclear. In contrast to kin-35 overexpression, which surprisingly did not cause lifespan extension (!), biochemical and genetic investigations of the kinase activity are necessary to provide mechanistic novelty for justifying publication in Nat. comm.

We thank the reviewer for pointing this out and we agree that, among others, the regulation and function of KIN-35 is a major question raised by our manuscript. To further emphasize this, we have expanded the discussion section of the paper.

Reviewer #2 (Remarks to the Author):

The authors have done a good job at addressing the issues raised on the initial version of their manuscript. The paper is now much stronger and I have no further comment on the revised manuscript.

We thank the reviewer for this positive assessment and are glad to make the findings available to the community now. Again, we want to thank the reviewer for the constructive assessment of the initial version of the manuscript.

Reviewer #3 (Remarks to the Author):

The authors have adequately responded to my comments.

We thank the reviewer for the insightful comments in the first round of revisions and are glad to see that our work has adequately addressed these important points.